# Regulation of pDC fate determination by histone deacetylase 3

Yijun Zhang[1,2†], Tao Wu[1,2†], Zhimin He[1,2†], Wenlong Lai[1,2], Xiangyi Shen[1,2], Jiaoyan Lv[1,2], Yuanhao Wang[1,2], Li Wu[1,2]*

[1]Institute for Immunology, Tsinghua-Peking Center for Life Sciences, School of Medicine, Tsinghua University, Beijing, China; [2]Beijing Key Laboratory for Immunological Research on Chronic Diseases, Beijing, China

**Abstract** Dendritic cells (DCs), the key antigen-presenting cells, are primary regulators of immune responses. Transcriptional regulation of DC development had been one of the major research interests in DC biology; however, the epigenetic regulatory mechanisms during DC development remains unclear. Here, we report that *Histone deacetylase 3* (*Hdac3*), an important epigenetic regulator, is highly expressed in pDCs, and its deficiency profoundly impaired the development of pDCs. Significant disturbance of homeostasis of hematopoietic progenitors was also observed in HDAC3-deficient mice, manifested by altered cell numbers of these progenitors and defective differentiation potentials for pDCs. Using the in vitro Flt3L supplemented DC culture system, we further demonstrated that HDAC3 was required for the differentiation of pDCs from progenitors at all developmental stages. Mechanistically, HDAC3 deficiency resulted in enhanced expression of cDC1-associated genes, owing to markedly elevated H3K27 acetylation (H3K27ac) at these gene sites in BM pDCs. In contrast, the expression of pDC-associated genes was significantly downregulated, leading to defective pDC differentiation.

*For correspondence:
wuli@tsinghua.edu.cn

†These authors contributed equally to this work

Competing interest: The authors declare that no competing interests exist.

## Editor's evaluation

This valuable study examines the expression of the histone-modifying enzyme HDAC3 within the dendritic cell (DC) compartment by taking advantage of a tamoxifen-inducible ERT2-cre mouse model to report the dependency of plasmacytoid DCs (pDCs) but not conventional DC (cDCs)s on HDAC3 at the common lymphoid progenitor stage during DC development. The methods are convincing and include RNA seq studies that identify multiple DC-specific target genes within the remaining pDCs, and the use of Cut and Tag technology to validate some of the identified targets of HDAC3. Taken together, this study shows the requirement of HDAC3 for pDCs but not for cDCs, congruent with the recent findings of a lymphoid origin of pDCs, and will be of great interest to immunologists.

## Introduction

Dendritic cells (DCs) are essential regulators of immune responses. Major DC subsets include conventional dendritic cells (cDCs) and plasmacytoid dendritic cells (pDCs). cDCs are professional antigen-presenting cells that are critical for initiation of antigen specific adaptive immunity and induction of immune tolerance. pDCs are the major producers of large amounts of type I interferon (IFN) during viral infection (*Reizis, 2019*). In the past decades, the knowledge on the cytokines, transcription factors, and progenitors involved in DC development had been obtained (*Nutt and Chopin, 2020*; *Anderson et al., 2021*). However, the role of epigenetic regulation in DC lineage determination and differentiation remains elusive.

The ultimate origin of DCs is the hematopoietic stem cells (HSCs), which give rise to multipotent progenitors (MPPs) with potential to populate entire hematopoietic system. MPPs produce common lymphoid progenitors (CLPs) and common myeloid progenitors (CMPs), which give rise to lymphoid and myeloid lineage cells, respectively. Both CLPs and CMPs have the potential to differentiate into DCs (*Sathe et al., 2013*; *Wu et al., 2001*; *D'Amico and Wu, 2003*; *Chicha et al., 2004*; *Manz et al., 2001*). CMPs further develop into common DC precursors (CDPs) with restricted potential for DC lineage differentiation (*Naik et al., 2007*; *Onai et al., 2007*), CDPs differentiate into pDCs and pre-cDCs in the bone marrow (*Liu and Nussenzweig, 2010*; *Puhr et al., 2015*). The CD115⁻CDP subsets predominantly produce pDCs, while CD115⁺CDP subsets produce more cDCs (*Onai et al., 2013*). Additionally, the Siglec H⁺Ly6D⁺ subset of LPs (CLPs) exhibits a specific potential to differentiate towards pDCs (*Rodrigues et al., 2018*; *Dress et al., 2019*).

Several transcription factors and key regulators have been identified to play important roles in pDC development. Transcription factor TCF4 (E2-2) is highly expressed in pDCs (*Nagasawa et al., 2008*). Deletion of TCF4 disrupts the development and function of pDCs (*Cisse et al., 2008*; *Ghosh et al., 2010*). The development of pDCs also requires PU.1, MTG16, ZEB2, and BCL11A, while expression of ID2 should be repressed (*Schlitzer et al., 2011*; *Carotta et al., 2010*; *Ghosh et al., 2014*; *Scott et al., 2016*; *Wu et al., 2016*; *Ippolito et al., 2014*; *Wu et al., 2013*; *Luc et al., 2016*; *Nagasawa et al., 2008*).

Histone deacetylase 3 (HDAC3) is a key epigenetic regulator orchestrating histone modification and chromatin remodeling (*Emmett and Lazar, 2019*). Previous studies show that HDAC3 regulates stem cell differentiation (*Summers et al., 2013*), T and B lymphocyte development (*Stengel et al., 2017*; *Stengel et al., 2019*; *Stengel et al., 2015*; *Philips et al., 2016*; *Hsu et al., 2015*), and the function of macrophages (*Chen et al., 2012*; *Nguyen et al., 2020*). Generation of DCs in vitro in the presence of pan-HDAC inhibitors reveals that HDAC family members are required for establishing a DC gene network (*Chauvistré et al., 2014*). However, it remains unknown which HDAC family member plays the major role in the development of DCs.

In this study, we observed a higher expression of *Hdac3* in pDCs compared to that in cDCs. Moreover, the development of pDCs was impaired in HDAC3-deficient mice, mainly due to the altered differentiation potential of pDCs in the HDAC3-deficient DC progenitors. Mechanistically, HDAC3 was required for repressing the H3K27ac level at cDC1-associated gene loci such as *Zfp366*, *Batf3* and *Zbtb46*, etc., thereby repressing the expression of cDC1-associated genes and promoting the differentiation towards pDCs. Together, our study revealed a crucial role of HDAC3 in regulating the development of pDCs from multiple hematopoietic progenitors with DC differentiation potential, and the histone deacetylase activity of HDAC3 is required for this process. Our findings provide novel insights into the epigenetic regulatory mechanisms for pDC development.

## Results
### HDAC3 deficiency resulted in defective development of pDCs

Genome-wide expression dataset ImmGen reveals that among class I HDACs, *Hdac1*, *Hdac2*, and *Hdac3* but not *Hdac8* are highly expressed by most immune cells. *Hdac3* is highly expressed in hematopoietic progenitors such as HSCs and CLP, and pDCs in spleen (*Figure 1—figure supplement 1A*). Consistently, high expression of Hdac3 was noticed in pDC progenitors, including HSCs, MMP4, CLP and CDPs, and bone marrow pDCs by RNA sequence analysis (*Figure 1—figure supplement 1B*). Furthermore, *Hdac3* expression in sorted DC subsets analyzed by quantitative real-time PCR (qRT-PCR) showed that pDCs expressed significantly higher levels of *Hdac3* than cDCs (*Figure 1—figure supplement 1C*). Thus, we speculated that *Hdac3* might play important roles in pDC development or function.

To test the role of HDAC3 in pDC development, *Hdac3*ᶠˡ/ᶠˡ mice were crossed with *Rosa26*-CreERᵀ² mice to generate HDAC3 conditional knockout mice (*Figure 1—figure supplement 2A*). *Hdac3*ᶠˡ/ᶠˡ CreERᵀ² mice and *Hdac3*ᶠˡ/ᶠˡ controls mice were treated with Tamoxifen (intraperitoneal injection, i.p.) for 5 consecutive days and DC subsets were analyzed 7 days after the last tamoxifen administration (*Figure 1—figure supplement 2B*). HDAC3 could be efficiently deleted in *Hdac3*ᶠˡ/ᶠˡ CreERᵀ² mice (*Figure 1—figure supplement 2C–D*). Murine pDCs express intermediate levels of CD11c, high levels of B220, CD45RA, Ly6C, and can be defined by specific markers such as Siglec H and PDCA-1

(*Anderson et al., 2021*), as shown in *Figure 1—figure supplement 3*. Compared to *Hdac3*[fl/fl] mice, *Hdac3*[fl/fl] CreER[T2] mice showed substantially reduced pDCs and the absolute number decreased about fourfold in BM and spleens (*Figure 1A and B*). CD11c[hi] cDCs were also slightly decreased, especially the CD24[+]CD172α[−] cDC1s, whereas the absolute numbers of CD24[−]CD172α[+] cDC2s were comparable (*Figure 1A and B*). To further confirm the number of pDC was reduced, additional pDC surface markers PDCA-1, Ly6C (*Figure 1C*), B220 and Ly6D (*Figure 1—figure supplement 4*) were also applied to identify pDC population, and all showed consistent reduction in pDC numbers in BM and spleen of *Hdac3*[fl/fl] CreER[T2] mice. pDCs are the major producers of type-I IFN, consistent with the reduction of pDCs in BM and spleen of *Hdac3*[fl/fl] CreER[T2] mice, the production of IFN-α by *Hdac3*[fl/fl] CreER[T2] bone marrow cells, splenocytes and splenic DCs were also decreased in response to type A CpG ODN stimulation in vitro (*Figure 1D*). Splenic cDC1 can also be identified using cell surface markers including XCR1 and CD8α. As shown in *Figure 1E–G*, in accordance with cDC1 identified by CD24, the percentage of splenic cDC1 marked by XCR1 showed little difference in *Hdac3*[fl/fl] CreER[T2] mice compared with that of *Hdac3*[fl/fl] mice. However, the expression of CD8α significantly decreased in HDAC3-deficient splenic cDC1 cells. Taken together, these results demonstrated the requirement of HDAC3 in pDC development in vivo.

## HDAC3 regulated pDC development in a cell-intrinsic manner

To test whether the function of HDAC3 in pDC development is cell-intrinsic, competitive BM chimeric mice were generated with BM cells from CD45.2[+] *Hdac3*[fl/fl] or *Hdac3*[fl/fl] CreER[T2] mice mixed with CD45.1[+] WT BM cells at 1:1 ratio. Eight weeks later, the chimeric mice were treated with Tamoxifen and the CD45.2[+] donor derived DC subsets were analyzed by flow cytometry. The percentage of *Hdac3*[fl/fl] CreER[T2] BM-derived CD45.2[+] cells were significantly lower than those derived from *Hdac3*[fl/fl] BM, indicating a competitive disadvantage of the HDAC3-deficient BM cells (*Figure 2A*). The percentage of CD45.2[+] cells derived from *Hdac3*[fl/fl] CreER[T2] donors only represented about 20% in BM and 30% in spleen of the chimeric mice (*Figure 2B*). Furthermore, the frequency of *Hdac3*[fl/fl] CreER[T2] derived BM and splenic pDCs among CD45.2[+] cells also decreased markedly, whereas the frequency of splenic cDC subsets were not significantly disturbed (*Figure 2A–C*). Moreover, when the BM cells were cultured in vitro for 5, 7, and 9 days in the presence of FLT3L and 4-hydroxytamoxifen (4-OHT) to induce *Hdac3* deletion in *Hdac3*[fl/fl] CreER[T2] BM cells, impaired pDC development from *Hdac3* deleted BM cells was also observed, in line with the results from in vivo study (*Figure 2D, E*). Meanwhile, cDC1 showed no significant change in percentage, despite of the decrease of cDC2 (*Figure 2D, F*). These data suggested that HDAC3 regulated pDC development in a cell intrinsic way.

## HDAC3 deletion led to disturbed homeostasis of early hematopoietic progenitors

To further investigate whether the absence of pDC in HDAC3-deficient mice was mainly due to decreased numbers of hematopoietic progenitors or resulted from inability of HDAC3-deficient progenitors to differentiate into pDCs, we first examined the composition of BM progenitor populations. BM Lin[−] cells from *Hdac3*[fl/fl] control or *Hdac3*[fl/fl] CreER[T2] mice after Tamoxifen treatment were analyzed for each progenitor populations. Compared to control mice, the Lin[−]Sca-1[+]CD117[+] (LSK) cells were increased significantly in *Hdac3*[fl/fl] CreER[T2] mice. Further analysis of LSK subpopulations (*Klein et al., 2022*; *Pietras et al., 2015*), we found that CD150[−]CD48[−]Flt3[−] long-term HSC (LT-HSC) and CD150[+] CD48[−]Flt3[−] short-term HSC (ST-HSC) were decreased, CD150[+]CD48[+]Flt3[−] MPP2 and CD150[−]CD48[+]Flt3[−] MPP3 were dramatically increased, while the CD150[−]Flt3[+] MMP4 was not significantly changed (*Figure 3A–C*). Meanwhile, HDAC3 deficiency led to markedly decreased number of Sca-1[−]CD117[+]CD16/32[int]CD34[+] CMP, while increased number of Sca-1[−]CD117[+]CD16/32[+]CD34[+] granulocyte-macrophage progenitor (GMP), but comparable number of Sca-1[−]CD117[+]CD16/32[−]CD34[−] megakaryocyte-erythroid progenitor (MEP) (*Figure 3D–F*). The major pDC precursor populations are the Sca-1[int]CD117[int]Flt3[+]CD127[+] CLPs and the CD117[int]Flt3[+]CD127[−]CD11c[−] CDPs which include both CD115[+] and CD115[−] subsets, whereas cDCs are mainly produced by CD115[+]CDP (*Rodrigues et al., 2018*; *Dress et al., 2019*; *Onai et al., 2007*; *Onai et al., 2013*). HDAC3-deficient mice showed a significant reduction in the numbers of CD115[+]CDP (*Figure 3E–G*) and CLP (*Figure 3A–C*), unexpectedly, the number of CD115[−]CDP cells were not significantly changed (*Figure 3G*). When further analyzed the pre-DCs (*Durai et al., 2019*; *Schlitzer et al., 2015*), we found dramatically reduced

Cell Biology | Immunology and Inflammation

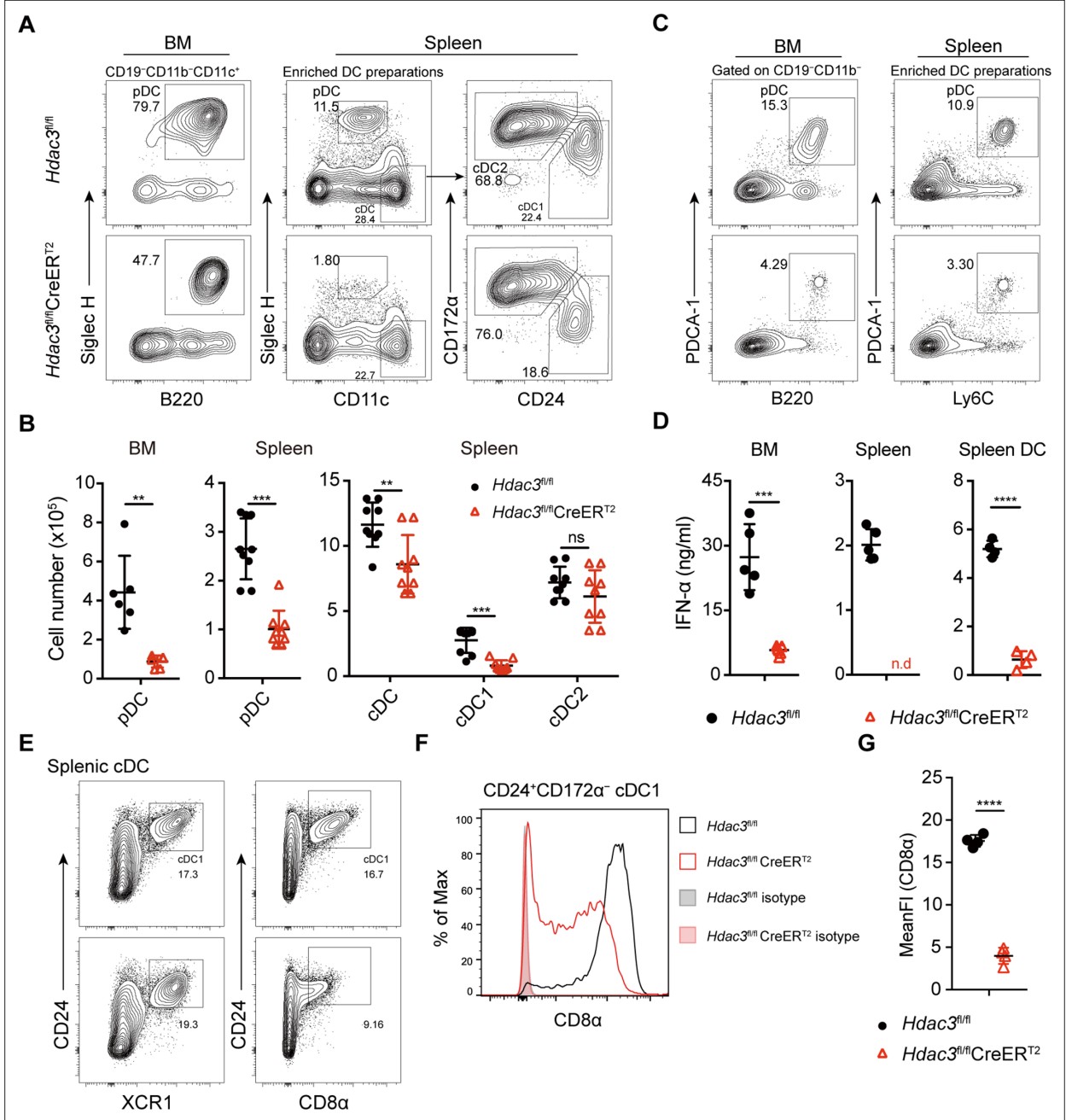

**Figure 1.** Impaired pDC development in *Hdac3*-deficient mice. (**A–B**) Representative flow cytometry profiles of BM pDCs and splenic DC subsets (**A**) and the absolute numbers of each DC subset of *Hdac3*^fl/fl and *Hdac3*^fl/fl CreER^T2 mice (**B**). (**C**) The pDC populations in BM and spleen were defined using a combination of indicated markers. Shown are staining profiles of gated CD11b⁻CD19⁻ cells from the BM and DC enrichment preparations from the spleens. (**D**) IFN-α production by *Hdac3*^fl/fl or *Hdac3*^fl/fl CreER^T2 cells in vitro. Total BM cells, splenocytes or purified DC preparations from the spleens were stimulated with CpG ODN 2216, and IFN-α in the supernatants was measured after 18 hr by ELISA. Cells were harvested from 4 to 5 mice each group. (**E**) The cDC1 populations in spleen were defined using a combination of indicated markers. Staining profiles of gated Siglec H⁻CD11c⁺ cells from the spleen are shown. (**F–G**) The expression and mean fluorescent intensities (MFIs) of CD8α in splenic CD24⁺CD172α⁻ cDC1s from *Hdac3*^fl/fl and *Hdac3*^fl/fl CreER^T2 mice. Data were pooled from two to three independent experiments, and shown as mean ± SD. * p<0.05; ** p<0.01; *** p<0.001; **** p<0.0001, by two-tailed Student's *t*-test.

The online version of this article includes the following source data and figure supplement(s) for figure 1:

**Figure supplement 1.** *Hdac3* was preferentially expressed in pDC.

**Figure supplement 2.** *Hdac3* was efficiently deleted by Tamoxifen treatment in *Hdac3*^fl/fl CreER^T2 mice.

**Figure supplement 2—source data 1.** Western blot of HDAC3 knockout efficiency in *Rosa26*-CreERT2 induce HDAC3 conditional knockout splenic DC

*Figure 1 continued*

subsets.

**Figure supplement 3.** Gating strategy for pDCs and cDCs.

**Figure supplement 4.** pDC gating by different surface markers.

pre-DC subsets, though the remaining pre-DCs were the cDC1-primed Siglec H⁻Ly6C⁻ population (*Figure 3—figure supplement 1*).

Taken together, apart from the increased number of LSKs, majorly the MPP2 and MPP3, and comparable number of CD115⁻CDPs, most of the other DC progenitors decreased significantly in HDAC3-deficient mice, as shown by the schematic diagram in *Figure 3H*.

It was reported that HDAC3 was required for the passage of HSCs through regulating S phase and the formation of early lymphoid progenitors (*Summers et al., 2013*). We then measured the DNA

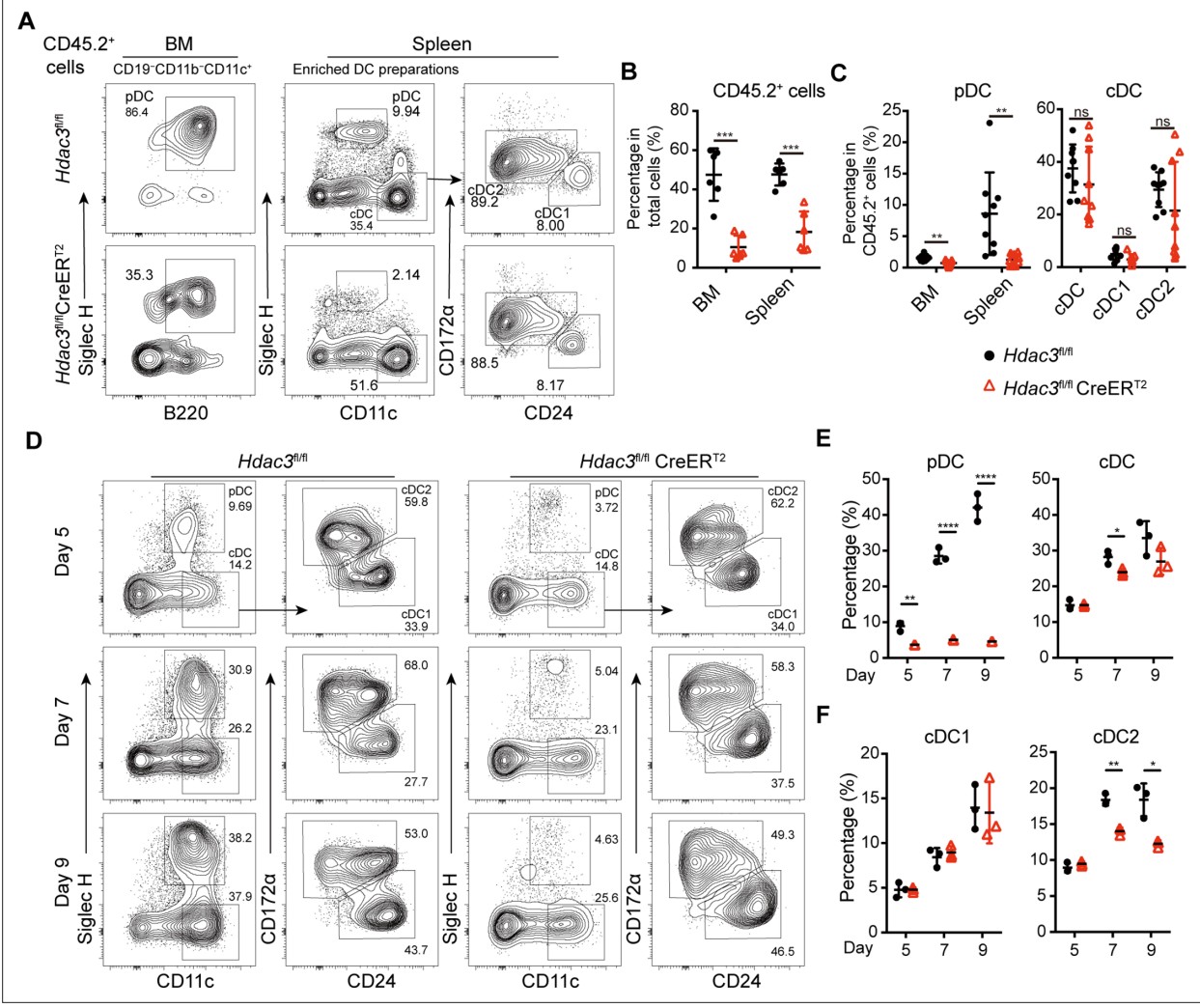

**Figure 2.** HDAC3 regulated pDC development in a cell-intrinsic manner. Lethally irradiated CD45.1 WT mice were reconstituted with a mixture of CD45.1 WT BM and BM from CD45.2 *Hdac3*ᶠˡ/ᶠˡ or *Hdac3*ᶠˡ/ᶠˡ CreERᵀ² mice at 1:1 ratio. Eight weeks after reconstitution, *Hdac3* deletion was induced by tamoxifen. (**A–C**) Results shown are staining profiles of gated CD11b⁻CD19⁻CD11c⁺ cells from the BM and enriched DC preparations from the spleens. Representative flow cytometry profiles of CD45.2⁺ donor-derived BM pDCs and splenic DC subsets (**A**), the percentage of CD45.2⁺ cells in BM and spleen (**B**), and the percentage of each DC subset among CD45.2⁺ cells (**C**) in the BM chimeric mice. Results are from one experiment representative of three independent experiments with three animals per group. (**D–F**) Total BM cells were plated at 1.5×10⁶ cells/ml in the presence of 200 ng/ml FLT3L and 1 µM 4-Hydroxytamoxifen (4OH-T). Results shown representative flow cytometry profiles (**D**) and the percentage of DC subsets (**E–F**) of FLT3L stimulated BM cultures on days 5, 7, and 9. Results are from one experiment representative of three independent experiments with three animals per group. Data are shown as mean ± SD. * p<0.05; ** p<0.01; *** p<0.001; **** p<0.0001, by two-tailed Student's *t*-test.

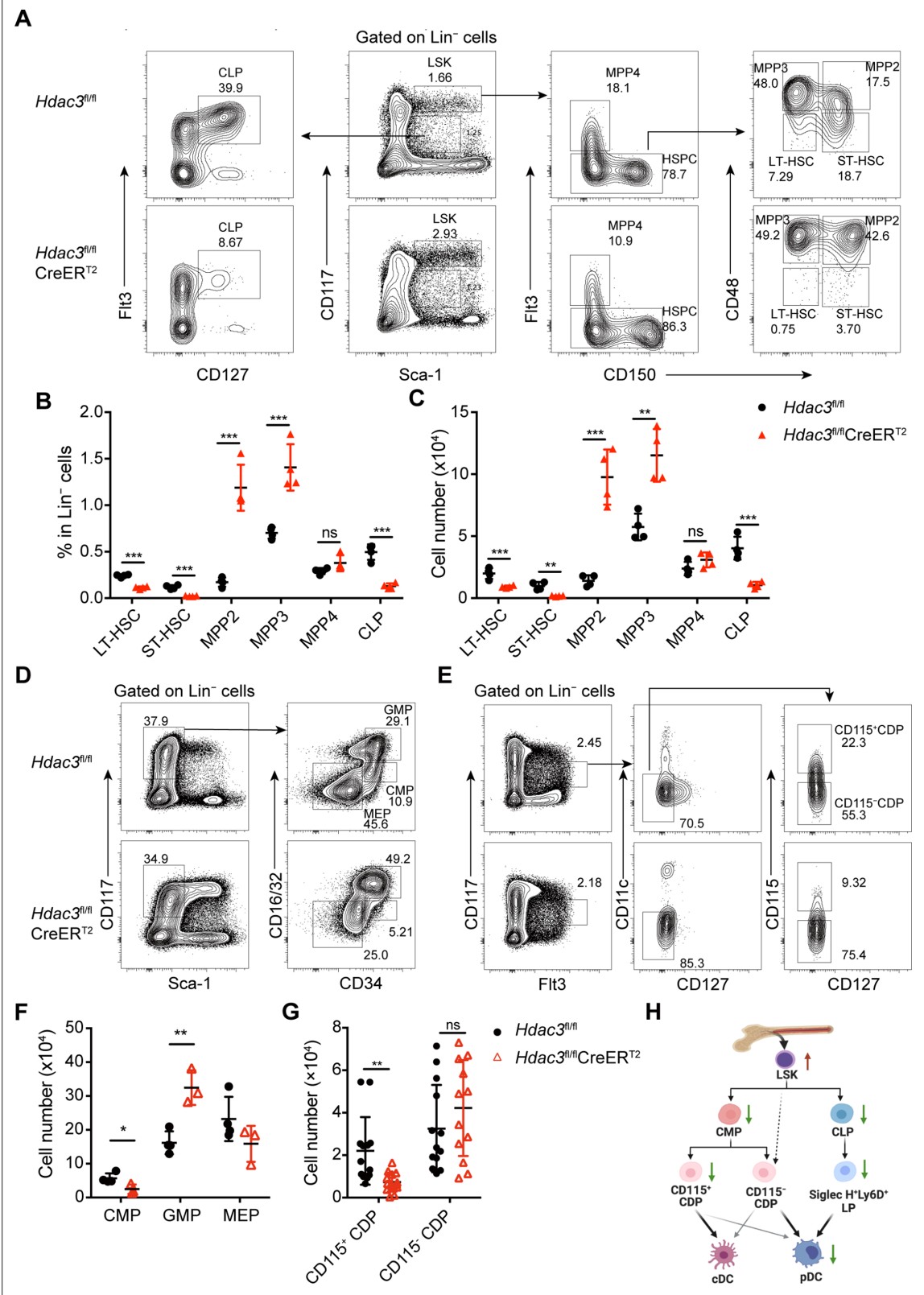

**Figure 3.** HDAC3 deficiency resulted in altered numbers of DC progenitor populations. (**A to C**) Representative flow cytometry profiles (**A**), the percentage (**B**) and the absolute cell number (**C**) of HSC subsets, MPP subsets and CLPs in *Hdac3*^fl/fl or *Hdac3*^fl/fl CreER^T2 mice.(**D–G**) Representative flow cytometry profiles (**D and E**) and the absolute cell number (**F and G**) of CMP, GMP and MEP (**D and F**), and CDP subsets (**E and G**) in *Hdac3*^fl/fl or *Hdac3*^fl/fl CreER^T2 mice. (**H**) A schematic diagram summarizing the changes in numbers of progenitors with DC differentiation potential in HDAC3

*Figure 3 continued on next page*

*Figure 3 continued*

deficient mice. Data were from one experiment representative of at least two independent experiments with three to four animals per group, and shown as mean ± SD. * p<0.05; ** p<0.01; *** p<0.001; **** p<0.0001, by two-tailed Student's *t*-test.

The online version of this article includes the following figure supplement(s) for figure 3:

**Figure supplement 1.** Pre-DC analysis in Hdac3 deficient mice.

**Figure supplement 2.** HDAC3 deficiency resulted in disturbed proliferation of DC progenitor cells.

synthesis in the DC progenitor cells during proliferation using BrdU-incorporation assay. Mice treated with Tamoxifen and then injected i.p. with BrdU 24 hr before analysis. Compared with that of control cells, HDAC3-deficiency led to an increased percentage of BrdU-incorporated LSKs, but decreased proportion of BrdU-incorporated CLPs, and comparable ratios of BrdU-incorporated CD115⁻CDPs and CD115⁺CDPs, respectively (*Figure 3—figure supplement 2A and B*). These results suggested that the disturbed homeostasis of progenitor cells caused by HDAC3 deficiency might mainly due to their altered proliferation rates.

Taken together, the disturbed homeostasis of BM progenitor cells might contribute to the defective pDC development in HDAC3-deficient mice. It remained unclear whether the differentiation potentials for pDCs of these progenitors were also affected by HDAC3 deficiency.

## HDAC3 deficiency abrogated pDC differentiation potential of LSKs and CD115⁻CDPs

As described above, HDAC3-deficient mice exhibited increased numbers of LSK cells and CD115⁻CDPs. To examine whether these increased progenitors could still give rise to pDCs, LSK cells and CD115⁻CDPs were purified from the BM of Tamoxifen treated CD45.2 *Hdac3*^fl/fl^ or *Hdac3*^fl/fl^ CreER^T2^ mice and then transplanted into lethally radiated CD45.1 recipient mice together with CD45.1 total BM competitors (*Figure 4A*). HDAC3-deficient LSKs and CD115⁻CDPs showed significant competitive disadvantage, revealed by the very low percentage of CD45.2⁺ cells in chimeric mice generated with HDAC3-deficient progenitors compared with that of *Hdac3*^fl/fl^ progenitors (*Figure 4B–C*). Furthermore, a decreased frequency of pDCs derived from HDAC3-deficient CD115⁻CDP was detected in both BM and spleen, whereas the frequencies of cDC subsets in the spleen were comparable to that derived from control CD115⁻CDP. These results indicated that HDAC3 deficiency in CD115⁻CDPs resulted in severe defect in pDC, but not in cDC development (*Figure 4D and E*). To further explore the regulatory role of HDAC3 in these progenitors, we compared gene expression profiles of LMPPs (the Flt3⁺CD34⁺ subset of LSKs) and CD115⁻CDPs from Tamoxifen treated *Hdac3*^fl/fl^ and *Hdac3*^fl/fl^ CreER^T2^ mice by RNA-seq analysis (*Figure 4—figure supplement 1*). HDAC3 deficiency resulted in down-regulation of 2533 and up-regulation of 206 genes with annotations in LMPPs (*Figure 4F*). KEGG pathway analysis of differentially expressed genes revealed a significant enrichment of genes for hematopoietic lineage pathway (*Figure 4G*), suggesting that HDAC3 may play important roles in hematopoietic linage cell differentiation. Interestingly, the HDAC3-deficient CD115⁻CDPs showed down regulated expression of genes that are highly expressed in pDC, such as *Siglech*, *Ly6c2*, *Ly6d*, *Cd209a*, and *Cox6a2* (*Figure 4H*). In addition, myeloid-related genes such as *Mpo*, *Ctsg*, and *Ms4a3* were up-regulated, but pDC-associated genes such as *Siglech*, *Ly6d*, and *Ly6c2* were down-regulated in both LMPP and CD115⁻CDPs (*Figure 4I*). These results indicated that HDAC3 is crucial for the expression of pDC-associated genes in those progenitors. Overall, these results indicated that HDAC3-deficient LSKs and CD115⁻CDPs were defective to differentiate into pDCs even though with increased cell numbers.

## HDAC3 was required for pDC development from multiple early hematopoietic progenitors

To further clarify whether HDAC3 is required for pDC development from hematopoietic progenitors at different developmental stages, including Flt3⁺LSKs (LMPPs), Flt3⁺CMPs, CLPs and CDPs, these progenitors from *Hdac3*^fl/fl^ or *Hdac3*^fl/fl^ CreER^T2^ mice (CD45.2⁺) were purified and cultured with CD45.1 BM feeders in vitro in the presence of FLT3L and 4OH-T. Both pDCs and cDCs generated from HDAC3-deficient LMPPs decreased significantly. Whereas HDAC3-deficient Flt3⁺CMPs, CLPs, CD115⁺CDPs and CD115⁻CDPs all showed profound defective pDC development in vitro, with no or

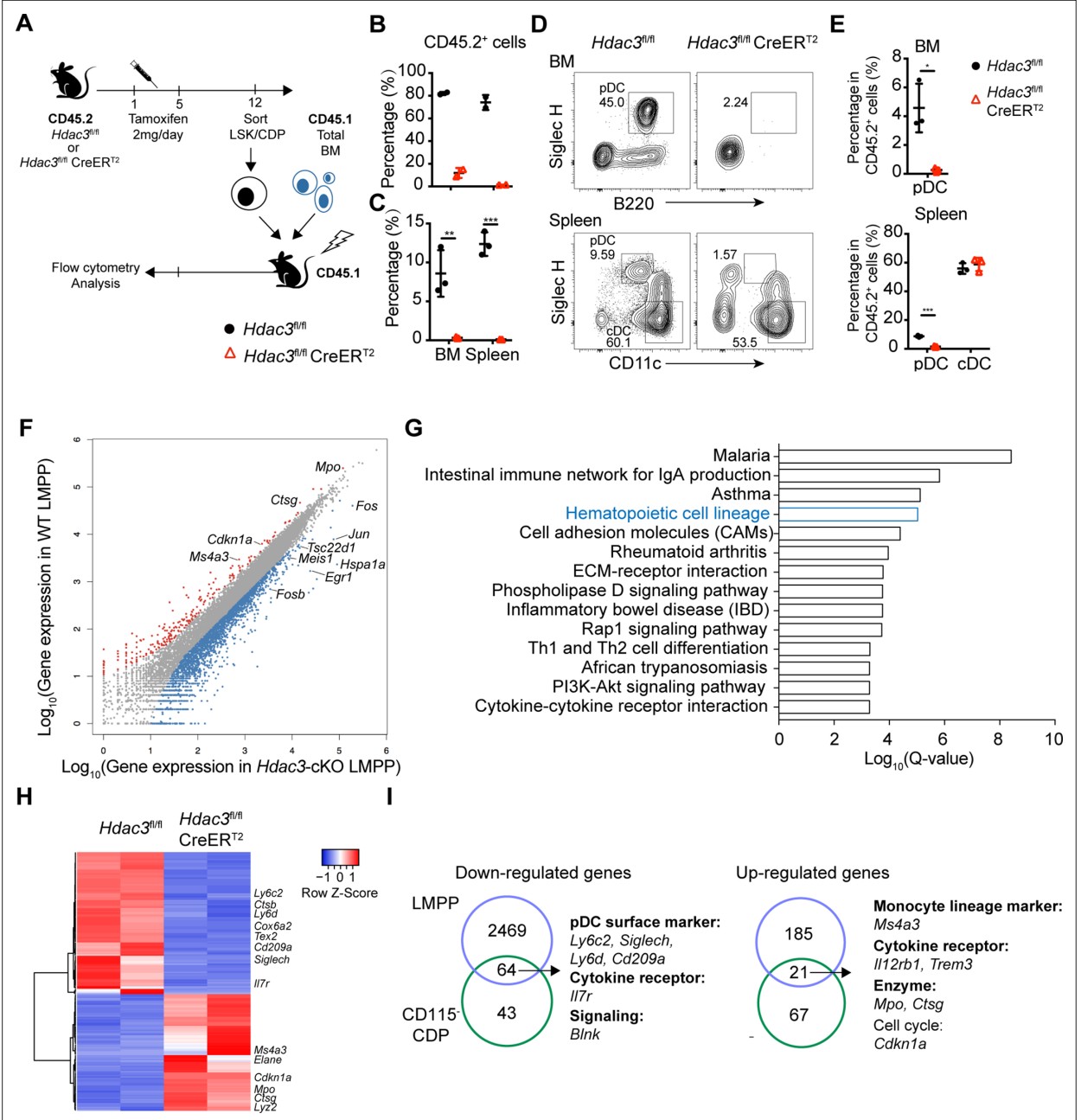

**Figure 4.** HDAC3 regulated pDC development by modulating the expression of pDC signature genes. (**A**) Schematic diagram of progenitor transplantation experiment. HSCs and CD115⁻CDPs were purified from the BM of *Hdac3*^fl/fl and *Hdac3*^fl/fl CreER^T2 mice after tamoxifen treatment, and then transplanted together with CD45.1 BM competitors into lethally irradiated CD45.1 WT mice. Three weeks (LSK) or 10 days (CD115⁻CDPs) post reconstitution, BM and spleen cells from BM chimeric mice were analyzed for pDC and cDC repopulation. (**B–C**) The percentage of CD45.2⁺ cells in BM chimeric mice repopulated with LSK (**B**) or CD115⁻CDP (**C**). (**D–E**) Representative flow cytometry profiles of BM pDCs and splenic DC subsets in CD45.2⁺ cells (**D**) and the percentage of DC subsets among CD45.2⁺ cells in the BM and spleen (**E**) of BM chimeric mice reconstituted with CD115⁻CDPs. Results are from one experiment representative of at least three independent experiments with two to three animals per group. Data are shown as mean ± SD. * p<0.05; ** p<0.01; *** p<0.001; **** p<0.0001, by two-tailed Student's *t*-test. (**F**) Scatter plot shows the differentially expressed genes (DEGs; Fold Change >2, q-value ≤0.001), which were up-regulated (red) and down-regulated (blue) in *Hdac3*-deficient LMPPs compared with WT LMPPs. Duplicate samples for each genotype were analyzed. (**G**) KEGG pathway analysis of DEGs in *Hdac3*-deficient LMPPs compared with WT LMPPs. Shown are pathways with Q-value ≤0.001. (**H**) Heatmap shows the DEGs (Fold Change >2, q-value ≤0.05) between *Hdac3*-deficient and WT CD115⁻CDPs. Duplicate samples for each genotype were analyzed. (**I**) Common genes down-regulated or up-regulated in *Hdac3*-deficient LMPPs and CD115⁻CDPs compared with those from WT LMPP and CD115⁻CDPs.

The online version of this article includes the following figure supplement(s) for figure 4:

**Figure supplement 1.** Purity of CD115⁻CDPs after sorting.

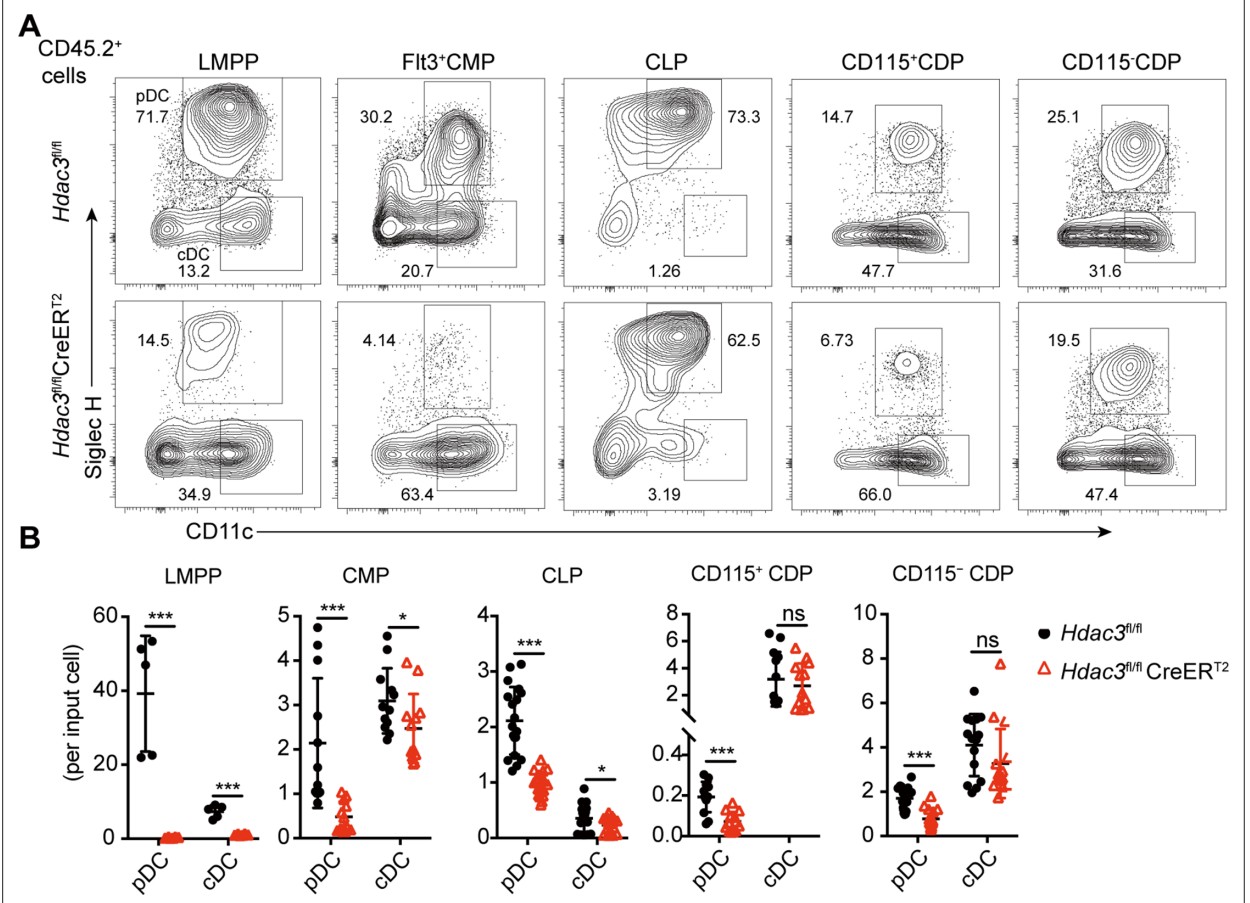

**Figure 5.** The development of pDCs from BM progenitors required HDAC3. The progenitor populations Flt3[+] HSCs, Flt3[+] CMPs, CLPs, and CDPs, were isolated from BM of *Hdac3*[fl/fl] or *Hdac3*[fl/fl] CreER[T2] mice, and cocultured with CD45.1 BM feeder cells in the presence of FLT3L with 1 µM 4OH-T. Representative flow cytometry profiles (**A**) and the number of DCs generated in each FLT3L stimulated culture per indicated input progenitor (**B**). Data were pooled from two to three independent experiments, and shown as mean ± SD. * p<0.05; ** p<0.01; *** p<0.001; **** p<0.0001, by two-tailed Student's *t*-test.

The online version of this article includes the following source data and figure supplement(s) for figure 5:

**Figure supplement 1.** HDAC3 was dispensable for Terminal differentiation of DCs.

**Figure supplement 1—source data 1.** Western blot of HDAC3 knockout efficiency in Itgax-Cre induce HDAC3 conditional knockout splenic DC subsets.

slight reduction in cDC development (*Figure 5A–B*). We further analyzed the *Hdac3*[fl/fl] *Itgax*-Cre mice, in which *Hdac3* was selectively deleted after CDP stage in CD11c[+] cells (*Figure 5—figure supplement 1A and B*, *Figure 5—figure supplement 1—source data 1*), to evaluate whether a later stage deletion of *Hdac3* would affect DC development in vivo. As shown in *Figure 5—figure supplement 1C, D*, pDCs and cDCs could develop normally in the spleen of *Hdac3*[fl/fl] *Itgax*-Cre mice.

Taken together, these results demonstrated that HDAC3 was required for pDC development from all early progenitors with DC differentiation potential, but dispensable for the later pDC differentiation.

## Deacetylase activity of HDAC3 was essential for its role in regulating pDC development

HDAC3 has been reported to be a dichotomous transcriptional activator and repressor during the activation of macrophages by LPS, with a non-canonical deacetylase-independent function to activate gene expression (*Nguyen et al., 2020*). To determine whether the deacetylase activity of HDAC3 is important for its role in regulating pDC development, the HDAC3 selective inhibitor RGFP966 (*Malvaez et al., 2013*) was added into FLT3L stimulated BM culture system. At low

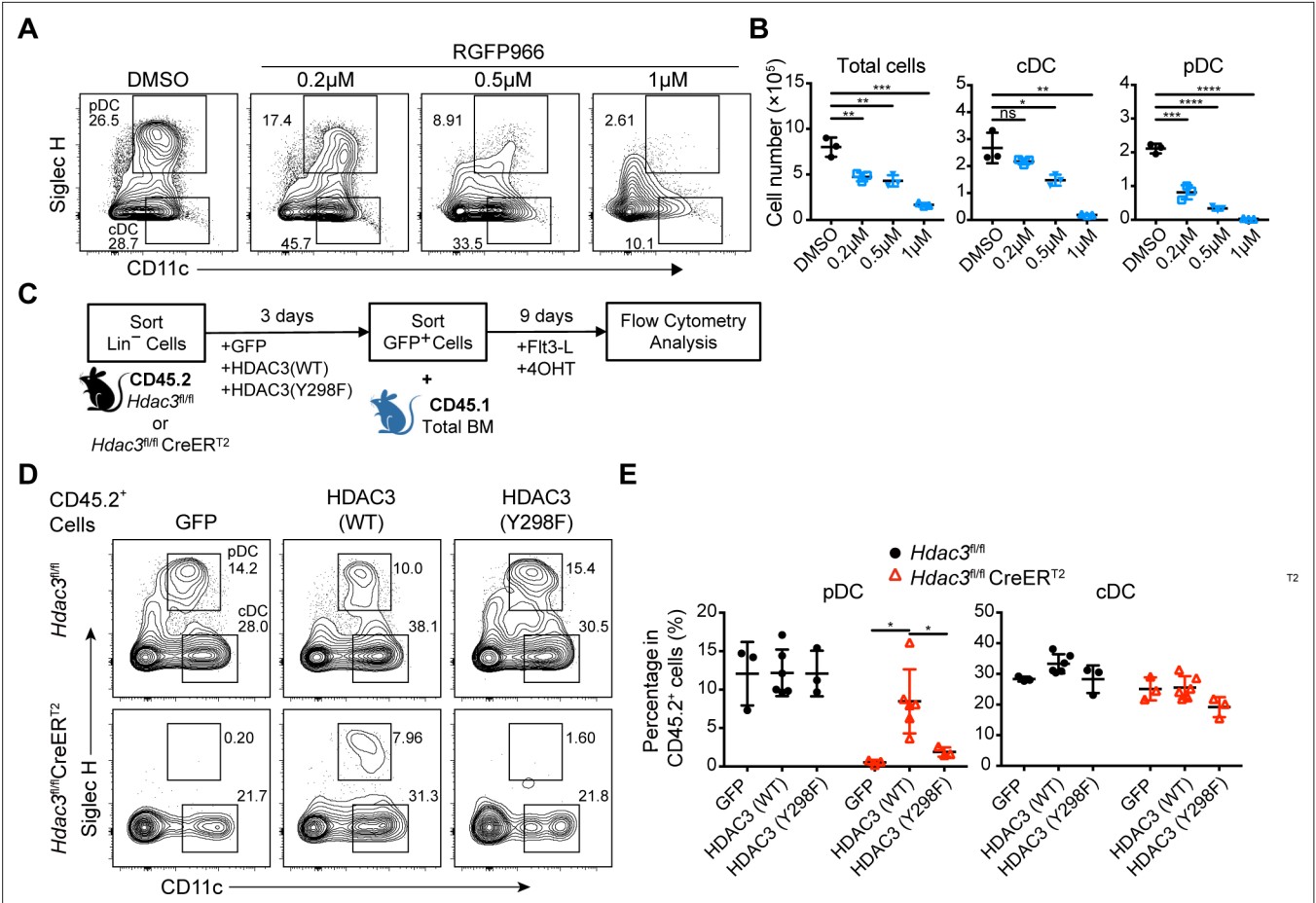

**Figure 6.** Deacetylase-dependent regulation of pDC development by HDAC3. (**A–B**) Total BM cells were plated at 1.5×10^6 cells/ml in the presence of 200 ng/ml FLT3L and indicated concentration of RGFP966, a selective HDAC3 inhibitor. Representative flow cytometry profiles (**A**), and the absolute number of total cells, pDC and cDC (**B**) generated in FLT3L stimulated BM cultures on day 9. Results are representative of at least 2 independent experiments with 3 mice per group. (**C**) BM Lin⁻ cells were transfected with empty (GFP), HDAC3-GFP (WT) and deacetylase-inactivated HDAC3-GFP (indicated as HDAC3(Y298F)) and cultured in the presence of FLT3L for 9 days. (**D–E**) Representative flow cytometry profiles (**D**), the percentage (**E**) of DCs generated in FLT3L stimulated cultures from retrovirus-transfected CD45.2⁺ BM Lin⁻ cells. Results are representative of at least two independent experiments with 3–6 duplicated wells per group. Data are shown as mean ± SD. * p<0.05; ** p<0.01; *** p<0.001; **** p<0.0001, by two-tailed Student's t-test.

concentration, RGFP966 could efficiently inhibit the differentiation of pDCs but not for cDCs. At higher concentration, it exhibited cellular toxicity, evidenced by the dramatic reduction in total cell numbers (*Figure 6A and B*). This result suggested that HDAC3 deacetylase activity was required preferentially for pDC development. The highly conserved tyrosine residue (Y298) in HDAC3 is located within the active site and is required for HDAC3 catalytic activity. HDAC3 (Y298F) mutant contains a tyrosine-to-phenylalanine substitution at residue 298, which inactivates HDAC3 deacetylase catalytic activity but not the interaction with NCoR/SMRT (*Sun et al., 2013*). The isolated Lin⁻ BM cells from *Hdac3*^fl/fl or *Hdac3*^fl/fl CreER^T2 mice were transduced with empty control vector (GFP), wild-type HDAC3 (WT) or HDAC3 (Y298F) mutant encoding retrovirus, to further confirmed the requirement of HDAC3 deacetylase activity in pDC development (*Figure 6C*). The amount of HDAC3 (Y298F) protein was comparable to those of HDAC3 (WT) in each group. The development of pDC from HDAC3-deficient Lin⁻ BM cells could be partially rescued by transducing the wild-type HDAC3. However, transduction of the deacetylase-inactivated HDAC3 (Y298F) mutant did not restore pDC development from HDAC3-deficient Lin⁻ BM cells (*Figure 6D and E*). Together, these results indicated that the deacetylase activity of HDAC3 is essential for its role in regulating the development of pDCs.

## HDAC3 regulated H3K27 acetylation level of cDC-associated genes

To explore the mechanism of HDAC3 regulation in pDC development, we compared gene expression profiles of BM pDCs from *Hdac3*[fl/fl] and *Hdac3*[fl/fl] CreER[T2] mice. Consistent with RNA-seq results obtained from progenitors, HDAC3-deficient pDCs showed enhanced enrichment of cell-cycle and cell proliferation process (*Figure 7—figure supplement 1* and *Supplementary file 1*), however, down-regulated pDC signature genes such as transcription factor *Tcf4* and pDC surface molecules *Siglech*, *Ly6c1*, *Ly6c2*, and *Ly6d* (*Figure 7A–D*). Among up-regulated genes in HDAC3-deficient pDCs, some genes were required in cDC1 development, such as *Spi1* (*Dakic et al., 2005*; *Carotta et al., 2010*; *Chopin et al., 2019*), *Zfp366* (*Zhang et al., 2021*), *Batf3* (*Hildner et al., 2008*; *Grajales-Reyes et al., 2015*) and *Id2* (*Jackson et al., 2011*; *Schlitzer et al., 2015*). Specific marker of cDC specification *Zbtb46* (*Meredith et al., 2012a*; *Meredith et al., 2012b*) and the surface molecule highly expressed by cDC1, such as *Clec9a* were also up-regulated in HDAC3-deficient pDCs. These results displayed a cDC-biased transcriptional profile in HDAC3-deficient pDCs.

HDAC3 associates with NCoR/SMRT to form co-repressor complex and represses gene expression by suppressing histone acetylation modifications. To examine whether the up-regulated genes in HDAC3-deficient pDCs were repressed by HDAC3, we explored H3K27ac level in BM pDCs from *Hdac3*[fl/fl] and *Hdac3*[fl/fl] CreER[T2] mice using CUT&Tag analysis. Motif analysis of unique H3K27ac peaks in HDAC3-deficient pDCs showed that these regions were significantly enriched with motifs recognized by PU.1 (SPI1) (*Figure 7—figure supplement 2A and B*). PU.1 promotes cDC development while repress pDC development by activating *Zfp366* expression (*Chopin et al., 2019*). HDAC3-deficient pDCs showed enhanced H3K27ac level around PU.1-binding region at cDC1-associated gene loci, including transcription factor *Zfp366*, *Zbtb46*, *Batf3* and cDC surface molecule *CD80*, *CD83* gene loci (*Figure 7E*, *Figure 7—figure supplement 2C*). The binding of HDAC3 on the gene loci of the transcription factors *Zfp366*, *Zbtb46*, *Batf3* were further confirmed by ChIP-qPCR analysis in BM pDCs. As shown in *Figure 7F*, significantly higher enrichments of HDAC3 on the sites of *Zfp366*, *Zbtb46*, *Batf3* loci than that of IgG control were observed. Since systemic Tamoxifen-induced *Hdac3* deletion in *Hdac3*[fl/fl] CreER[T2] mice led to death within 8~9 days, we generated BM chimeric mice by transplanting BM cells from untreated CD45.2 *Hdac3*[fl/fl] or *Hdac3*[fl/fl] CreER[T2] mice into lethally irradiated CD45.1 WT recipient. Eight weeks after reconstitution, recipient mice were treated with Tamoxifen, and DC subsets from BM chimeric mice were analyzed 3 weeks later to test the long-term DC differentiation ability of HDAC3-deficient progenitors (*Figure 7—figure supplement 2D*). Analysis of CD45.2[+] donor cells showed that pDC development from *Hdac3*[fl/fl] CreER[T2] BM cells were severely defective. Meanwhile, the percentage and absolute number of splenic cDC1 derived from *Hdac3*[fl/fl] CreER[T2] BM cells were increased, while that of cDC2 were decreased (*Figure 7G and H*). Taken together, these results suggested that HDAC3 could repress cDC1-associated gene expression through histone deacetylation and thereby promoted the expression of pDC-associated genes and subsequent pDC development.

## Discussion

In this study, we identified HDAC3 as a key epigenetic regulator for pDC development, evidenced by the absence of pDCs in BM and spleens of the HDAC3-deficient mice. Further investigation revealed that HDAC3 deficiency resulted in disturbed homeostasis of BM hematopoietic progenitors with DC differentiation potential. Moreover, we showed that HDAC3 was required for pDC differentiation from all early progenitors with DC differentiation potential. Mechanistic study by RNA-seq analysis comparing the gene expression profiles of the major progenitor populations and BM pDCs from HDAC3-deficient and control mice revealed significant downregulations of pDC-associated gene expression and upregulations of cDC1-associated genes. To further clarify how HDAC3 regulated the expression of these genes, we performed CUT&Tag analysis and ChIP-qPCR analysis. The results suggested that HDAC3 regulated the acetylation of H3K27 at cDC1-associated gene loci, thereby repressed the expression of cDC1-associated genes and promoted the expression of pDC-associated genes and subsequent differentiation of the progenitors towards pDCs. Consistently, HDAC3 deficiency resulted in decreased expression of pDC-associated genes and profound defect in pDC development. Thus, our study revealed a novel epigenetic regulatory role of HDAC3 in pDC development.

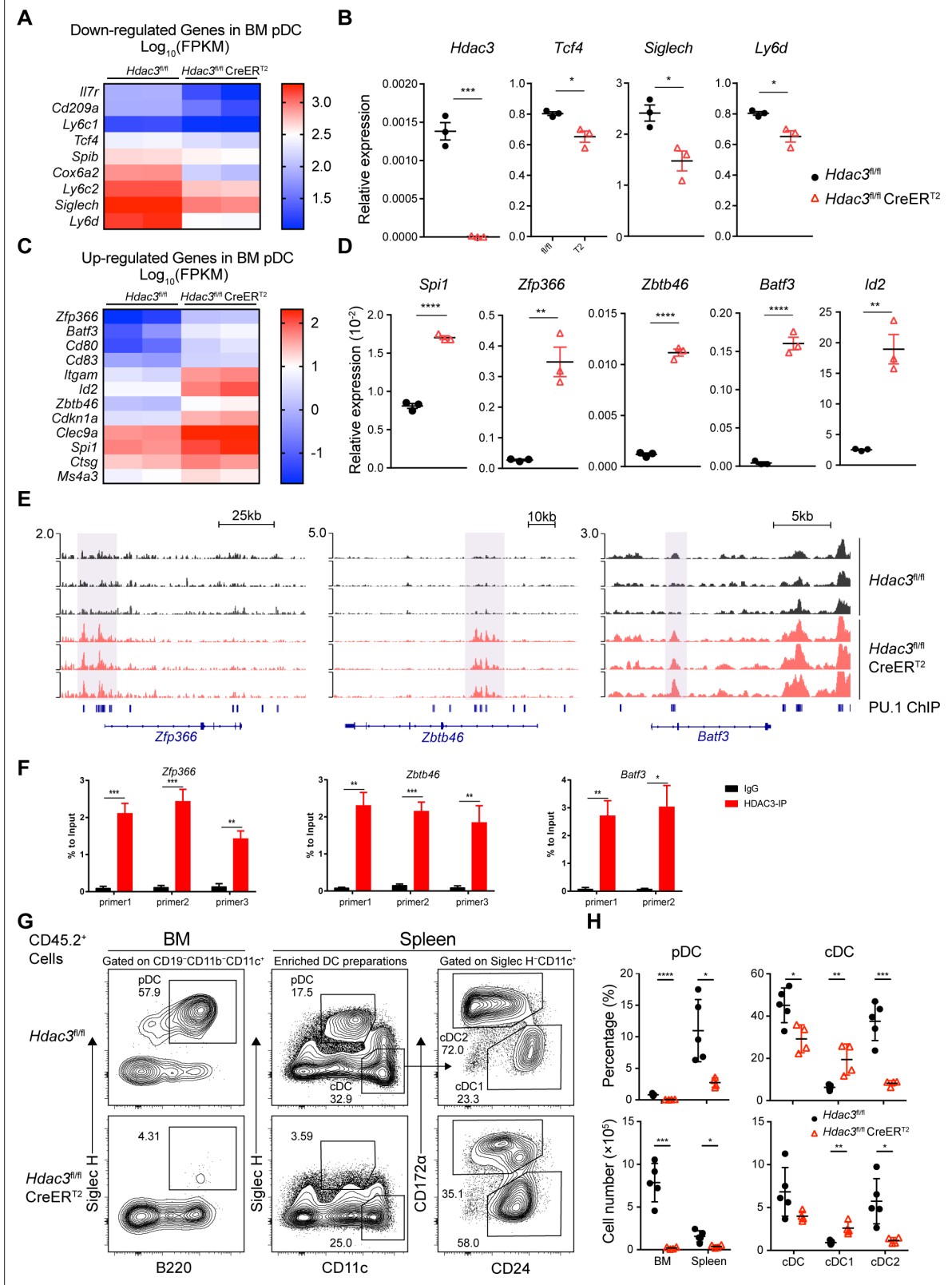

**Figure 7.** HDAC3 repressed H3K27ac around PU.1-binding sites at cDC genes loci. (**A–D**) Up-regulated and down-regulated DEGs (Fold Change >2, q-value ≤0.05) in *Hdac3*-deficient BM pDCs compared with WT pDCs, determined by RNA-seq analysis (**A, C**) and confirmed by qRT-PCR (**B, D**). (**E**) Sequencing tracks of CUT&Tag analysis, with anti-H3K27ac antibody, of BM pDC sorted from *Hdac3*^fl/fl and *Hdac3*^fl/fl CreER^T2 mice. Loci of cDC transcription factor *Zfp366*, *Zbtb46*, *Batf3* are displayed. (**F**) ChIP-qPCR validation of HDAC3 binding sites in BM pDCs on *Zfp366*, *Zbtb46*, *Batf3* loci

*Figure 7 continued on next page*

*Figure 7 continued*

identified in CUT&Tag analysis. Representative data of two independent experiments with similar pattern. (**G–H**) Lethally irradiated CD45.1 WT mice were reconstituted with BM alone from CD45.2 *Hdac3*$^{fl/fl}$ or *Hdac3*$^{fl/fl}$ CreER$^{T2}$ mice. Eight weeks after reconstitution, *Hdac3* deletion was induced by Tamoxifen and BM cells and splenocytes were analyzed after three weeks. Representative flow cytometry profiles of CD45.2$^+$ BM pDCs and splenic DC subsets (**G**), the percentage of DC subsets among CD45.2$^+$ cells and absolute cell number (**H**) in BM chimeric mice. The staining profiles of pDC and cDC subsets on gated CD45.2$^+$CD11b$^-$CD19$^-$CD11c$^+$ cells from the BM and CD45.2$^+$ enriched DC preparations from the spleens are shown. Results are from one experiment representative of three independent experiments with four to five animals each group. Data are shown as mean ± SD. * p<0.05; ** p<0.01; *** p<0.001; **** p<0.0001, by two-tailed Student's *t*-test.

The online version of this article includes the following figure supplement(s) for figure 7:

**Figure supplement 1.** HDAC3 deficiency increased the cell-cycle related genes.

**Figure supplement 2.** HDAC3 regulated H3K27ac around PU.1-binding sites at cDC genes loci.

The development of DCs from hematopoietic progenitors is precisely orchestrated. We observed that deletion of *Hdac3* resulted in marked decrease in the numbers of CMPs, CLPs and CD115$^+$CDPs, which partially accounted for the reduced pDC generation, whereas the numbers of HDAC3-deficient LSKs, especially MPP2 and MPP3, were significantly increased. However, despite the increase in the numbers, their abilities to differentiate into pDCs were severely impaired. The reduction in the number of CLP and increase in the number of LSKs in HDAC3-deficient mice were in line with the results of the BrdU incorporation analysis that showed a reduced DNA replication in HDAC3-deficient CLPs and an increased DNA synthesis in HDAC3-deficient LSKs. The incorporation level of BrdU in HDAC3-deficient and control CD115$^+$CDPs and CD115$^-$CDPs were comparable, though the number of HDAC3-deficient CD115$^+$CDPs was reduced. Considering that total number of CDPs did not change significantly, and a decrease of *Csf1r* (coding CD115) expression could be observed in RNA-seq analysis of the HDAC3-deficient CDPs. These results suggested that the reserved number of CD115$^-$CDPs might be due to the downregulation of *Csf1r* expression in CD115$^+$CDPs. Overall, these results suggested that HDAC3 regulated different hematopoietic progenitors with different regulator mechanisms.

Cell fate determination is a key process during cell development, in which transcription factors play important roles. Recent studies reported PU.1-DC-SCRIPT-IRF8 axis as an important transcriptional pathway that regulated the fate determination between pDC and cDC1. PU.1 can activate *Zfp366* (encodes DC-SCRIPT) expression and DC-SCRIPT enhanced the expression of *Irf8* and thereby promoted the development and function of cDC1 (*Zhang et al., 2021*). Loss of DC-SCRIPT leads to defective differentiation of cDC1 and an enhanced development of pDC (*Chopin et al., 2019*). These studies demonstrated that cell fate determination of cDC1 versus pDC was regulated in a mutually restrictive manner. In line with this, we also observed in the BM chimeric mice that HDAC3-deficient BM cells generated decreased number of pDCs but increased number of cDC1 compared to that derived from wildtype BM 3 weeks after Tamoxifen treatment, suggesting that HDAC3 may be involved in the fate determination of pDCs versus cDC1. This was confirmed by RNA-seq and qRT-PCR analysis of the residual HDAC3-deficient BM pDCs that showed down-regulation of pDC-signature genes such as *Tcf4*, *Siglech* and up-regulation of cDC1-associated genes, including *Zfp366*, *Id2* and *Batf3*, indicating that HDAC3 may promote pDC differentiation by repressing the expression of cDC1-associated genes. Further Cut & Tag analysis indicated that HDAC3-KO resulted in enhanced H3K27ac on *Zfp366*, *Zbtb46* and *Batf3* loci, and ChIP-qPCR analysis verified the binding of HDAC3 on these sites. Taken together, our study defined an essential role of HDAC3 in the regulation of pDC versus cDC1 lineage determination and revealed a new epigenetic regulatory mechanism for pDC development.

pDCs are the major producers of type I interferon and are involved in the development of some autoimmune diseases (*Reizis, 2019*). In this study we also found that HDAC3 specific inhibitor RGFP966 could efficiently block pDC generation in vitro, it therefore might serve as a potential therapeutic agent for pDC related diseases. Further investigation is warrant to evaluate this potential. Taken together, our study defined an essential role of HDAC3 in the regulation of pDC versus cDC lineage differentiation, and revealed a new epigenetic regulatory mechanism for pDC development.

# Materials and methods

## Mice

All mice used in this study were C57BL/6 background. B6N-Tyr^(c-Brd) *Hdac3*^tm1a(EUCOMM)Wtsi/Wtsi mice were gifts from Prof. Fang-Lin Sun (Tongji University), FLP-DELETER mice: B6.129S4-*Gt(ROSA)26Sor*^tm1(FLP1)Dym/RainJ (JAX:009086, gain from Biocytogen, Beijing, China) widely expressed of the FLPe variant of the *Saccharomyces cerevisiae* FLP1 recombinase gene driven by the Gt(ROSA)26Sor promoter. *Hdac3*^tm1a(EUCOMM)Wtsi/Wtsi were crossed with FLP-DELETER mice to remove the neomycin (Neo) cassette flanked by two FRT sites, thus generating *Hdac3*^flox/flox (*Hdac3*^fl/fl) with *Hdac3* exon 3 floxed by a pair of loxP site. *Itgax*-Cre mice were gifts from Prof. Nan Shen (Shanghai Jiao Tong University) and B6.129-*Gt(ROSA)26Sor*^tm1(cre/ERT2)Tyj/J (*Rosa26*-CreER^T2) mice were purchased from The Jackson Laboratory. *Hdac3*^fl/fl mice were crossed with *Itgax*-Cre or *Rosa26*-CreER^T2 mice to generate *Hdac3*^fl/fl *Itgax*-Cre mice and *Hdac3*^fl/fl CreER^T2 mice. Mice were bred and maintained in a specific pathogen-free (SPF) animal facility at Tsinghua University, with 12/12 hr light/dark cycle, at 22–26°C and sterile pellet food and water and libitum. All animal procedures were performed in strict accordance with the recommendations and approval of the Institutional Animal Care and Use Committee at Tsinghua University.

## Cell isolation and flow cytometry

For flow cytometry, BM cells and spleens were collected. Spleens were then minced into fine pieces and digested with Collagenase III (1 mg/ml, Worthington) and DNase I (0.1 mg/ml, Roche) in RPMI-1640 for 30 min at room temperature with rotating. Erythrocytes were removed by using Red Cell Removal Buffer (0.168 M NH4Cl). Cell suspensions were filtrated through a 70 µm cell strainer. BM cells or splenocytes were then separated by density gradient centrifugation over 1.086 g/mL or 1.077 g/mL Nycodenz respectively (Sigma-Aldrich). The cell fraction harvested from the interface between fetal bovine serum (FBS) and Nycodenz was then incubated with antibody cocktail. In this study, we used two different panels of antibody cocktails, one for Lin- cells, including mAbs against: CD2/CD3/TER-119/Ly6G/B220/CD11b/ CD8/CD19; the other for DC enrichment, including mAbs to CD3/CD90/TER-119/Ly6G/CD19. Cells were washed and incubated with BioMag Goat Anti-Rat IgG (Bangs Laboratories) on ice. Beads were then removed using magnetic separator. Cell suspensions were incubated with homemade anti-CD16/32 antibody or Rat Gamma Globulin (Jackson ImmunoResearch) for blocking and then stained with fluorochrome-conjugated monoclonal antibodies against CD45, CD11c, Siglec-H, B220, PDCA-1, CD45RA, Ly6C, CD8α, CD172α, XCR1, and CD24, and with 7-AAD or Flexible viability dye eFluor506 to distinguish and exclude dead cells. Cells were analyzed on the LSR Fortessa instrument (Becton Dickinson), and data were analyzed with FlowJo X software (Tree-Star). Cell sorting was performed with a FACS Aria III cell sorter (Becton Dickinson) for subsequent experiments. Antibodies are listed in *Table 1*.

**Table 1.** Reagents.

| Antibody | Clone | Source |
| --- | --- | --- |
| CD45 | 30-F11 | BioLegend |
| CD45.1 | A20 | BioLegend |
| CD45.2 | 104 | BioLegend |
| CD19 | EBio103 | eBioscience |
| B220 (CD45R) | RA3-6B2 | eBioscience |
| CD3ε | 145–2 c11 | eBioscience |
| CD4 | GK1.5 | eBioscience |
| CD8α | 53–6.7 | BioLegend |
| CD11b | M1/70 | eBioscience |
| CD11c | N418 | eBioscience |
| Siglec H | eBio440C | eBioscience |
| PDCA-1 (CD317) | 927 | BioLegend |
| XCR1 | ZET | BioLegend |
| CD205 | 205yekta | eBioscience |
| CD24 | M1/69 | BioLegend |
| CD127 (IL-7Rα) | A7R34 | BioLegend |
| CD117 (c-Kit) | ACK2 | BioLegend |
| CD115 (c-fms) | AFS98 | BioLegend |
| FLT3 (CD135) | A2F10 | eBioscience |
| Sca-1 (Ly-6A/E) | D7 | eBioscience |
| CD34 | RAM34 | eBioscience |
| CD16/32 | 93 | BioLegend |

## FLT3L stimulated bone marrow culture

DC generation by FLT3L stimulated BM cultures were performed based on the original protocol reported by *Naik et al., 2005*. Briefly, total BM cells were plated at 1.5×10⁶ cells/ml in RPMI-1640 complete medium, in the presence of 200 ng/ml FLT3 (PeproTech) and 1 µM 4-Hydroxytamoxifen (Sigma-Aldrich). Cells were harvested and analyzed on day 5–9. To generate DCs from progenitors in vitro, the progenitor populations LMPPs ($10^3$), Flt3⁺ CMPs, CLPs, and CDPs ($10^4$), were isolated and cocultured with 1.5×10⁴ CD45.1 BM feeder cells in the presence of 200 ng/ml FLT3L with 1 µM 4-Hydroxytamoxifen. Cells were harvested and analyzed on day 4 (for CLPs or CDPs), day 5 (for CMPs) or day 9 (for LMPPs).

## Bone marrow reconstitution

To generate mixed BM chimeras, 1×10⁶ CD45.2 *Hdac3*^fl/fl or *Hdac3*^fl/fl CreER^T2 bone marrow (BM) cells were mixed with CD45.1 wild type (WT) competitor BM cells at 1:1 ratio, and then transplanted into lethally irradiated CD45.1 WT mice. Eight weeks later, *Hdac3* deletion was induced by Tamoxifen treatment. To generate DCs from progenitors in vivo, 10⁴ LSKs or CD115⁻CDPs were sorted from *Hdac3*^fl/fl or *Hdac3*^fl/fl CreER^T2 mice after Tamoxifen treatment, and then were transplanted into lethally irradiated CD45.1 WT mice mixed with 2×10⁵ CD45.1 BM cells. Three weeks (for LSKs) or 10 days (for CD115⁻CDPs) post reconstitution, BM cells and enriched DC preparations were analyzed. To examine DC differentiation potential of HDAC3-deficient progenitors in the long-term, lethally irradiated CD45.1 WT mice were reconstituted with untreated BM of CD45.2 *Hdac3*^fl/fl or *Hdac3*^fl/fl CreER^T2 mice. Eight weeks later, *Hdac3* deletion was induced by Tamoxifen and BM cells and splenocytes were analyzed three weeks post treatment.

## Retrovirus production and infection

Retroviral supernatants with pMYS-iresGFP, pMYS-HDAC3(WT)-iresGFP, pMYS-HDAC3(Y298F)-iresGFP were centrifuged onto RetroNectin (Takara)-coated plates for 2 hr at 2000 *g* at 32 °C. Sorted BM Lin- cells were cultivated with the virus in the presence of IL-3, IL-6 and SCF for 48 hours. GFP + cells were then sorted.

## Cleavage under targets and tagmentation (Cut & Tag) analysis

The Hyperactive In-Situ ChIP Library Prep Kit was purchased from Vazyme, and libraries were generated following the manufacturer's instructions. Briefly, 1×10⁵ BM pDCs sorted from *Hdac3*^fl/fl control or *Hdac3*^fl/fl CreER^T2 littermates were bound to beads and were subjected for immunoprecipitation with primary anti-H3K27ac antibody (Active Motif) at 4 °C overnight. A secondary anti-rabbit antibody (Abcam) was then added and incubated under gentle agitation at room temperature (RT) for 1 hr. Cells were then washed and incubated in Dig-300 buffer with 0.04 µM hyperactive pG-Tn5 transposon at RT for 1 hr. Cells were washed and then incubated in 100 µl of tagmentation buffer (10 mM MgCl2 in Dig-300 buffer) at 37 °C for 1 hr. Reaction was stopped by addition of SDS, EDTA and proteinase K and incubated at 55 °C for 1 hr and then DNA were extracted. For library amplification, 24 µL of DNA was mixed with 10 µL 5×TruePrep Amplify Buffer (TAB), 1 µL TAE, and 5 µL barcoded i5 and i7 primers and amplified for 20 cycles. PCR products were purified with VANTS DNA Clean beads (Vazyme). Libraries were sequenced on an Illumina NovaSeq platform.

## RNA extraction and quantitative RT-PCR

RNA of sorted cells was extracted by TRIzol Reagent (Invitrogen), and cDNA was reversely transcribed using PrimeScript RT Master Mix (Takara). For quantitative RT-PCR, PowerUp SYBR Green Master Mix (Applied Biosystems) was used, and samples were run on an HT7900 (Applied Biosystems) quantitative PCR machine. Gene expression was normalized to the expression of house-keeping gene Actin. Primers used: *Hdac3*: (forward) 5'-TGATCGTCTTCAAGCCTTAC-3' (reverse) 5'-TTGGTGAAACCCTGCATAT-3', *Batf3*: (forward) 5'-GGAACCAGCCGCAGAG-3', (reverse) 5'-GCGCAGCACAGAGTTCTC-3', *Spi1*: (forward) 5'-ATGTTACAGGCGTGCAAAATGG-3', (reverse) 5'-TGATCGCTATGGCTTTCTCCA-3', Zfp366 (forward) 5'-CTTCCTGCCCAAGCAGCCCC-3' (reverse) 5'-GGCACTGCCAGCGCTTCTGA-3', *Id2* (forward) 5'-ACCAGAGACCTGGACAGAAC-3' (reverse) 5'-AAGCTCAGAAGGGAATTCAG-3', *Zbtb46*: (forward) 5'-GACACATGCGCTCACATACTG-3' (reverse) 5'-TGCACACGTACTTCTTGTCCT-3', *Tcf4*: (forward) 5'-CGAAAAGTTCCTCCGGGTTTG-3' (reverse) 5'-CGTA

GCCGGGCTGATTCAT-3', *Siglech*: (forward) 5'-ggaaccaacctcacctgtca-3' (reverse) 5'-agagacatgggct-gtggagt-3', *Ly6d*: (forward) 5'-cctcagcctgctcactgtta-3' (reverse) 5'-caagggaaattccaagcagt-3', *Actin*: (forward) 5'-ATGCTCCCCGGGCTGTAT-3' (reverse) 5'-CATAGGAGTCCTTCTGACCCATTC-3'

## Protein extraction and western blot analysis

Sorted DC subsets were lysed in cold RIPA Lysis Buffer (Beyotime Biotechnology) supplemented with protease inhibitor cocktail (Roche), phosphatase inhibitor cocktail (Bimake), and SDS loading buffer (Beyotime Biotechnology). Total protein was denatured and loaded into the SDS-PAGE, and then transferred to PVDF membranes. After blocking, primary antibodies were hybridized overnight (HDAC3, 10255–1-AP, ProteinTech; β-actin, 3700, CST) and then secondary antibodies. Imaging was carried out with Amersham Imager 600 System (GE).

## ChIP-qPCR analysis of HDAC3 binding sites in BM pDCs

ChIP-qPCR analysis were performed using formaldehyde fixed purified BM pDCs with ChIP-IT Express Chromatin Immunoprecipitation Kits (53035, Active Motif, USA) according to the manufacturer's instructions. Briefly, sorted BM pDCs were fixed using 1% formaldehyde for 10 min and quenched with 10×glycine in the kit. Then cells were washed with PBS for 3 times and resuspended using ChIP lysis in the kit. After centrifuged at 2000 rpm for 5 min, the cells were the lyzed in digest buffer with proteinase inhibitor cocktail and phosSTOP (Roche, Switzerland) on ice. Then the cells were sonicated using Bioruptor Pico sonication device (Diagenode, Belgium) with 30 s on/ 30 s off for 5–6 cycles at 4 °C, and then add enzymatic shearing cocktail to the system and incubated at 37 °C for 8 min, reaction was stopped by adding of EDTA and the cell lysate was centrifuged at 12,000 rpm for 10 min. Before diluted in ChIP dilution buffer (with proteinase inhibitor cocktail and phosSTOP) appropriate volume of supernatant was collected as 20% Input. Then the sample were divided into two equal volumes and antibody of HDAC3 (85057 S, CST, USA) or normal rabbit IgG (2729 S, CST, USA) was added according to the antibody instruction and incubated at 4 °C overnight. Then 10 µl of pre-washed Magna ChIP Protein A+G Magnetic Beads (16–663, Millipore, USA) were added to each sample respectively and incubated for 2.5 hr. Then the beads were washed sequentially with low salt buffer, high salt buffer and TE buffer at 4 °C, the beads were then washed by TE buffer again and then remove the entire supernatant. Then the beads sample and input were added 172 µl TE buffer, 10 µl 10% SDS and 8 µl 5 M NaCl, and incubated at 65 °C overnight shaking vigorously. RNase A in the kit were then added and incubated at 37 °C for 15–30 min followed by adding the proteinase K (20 mg/ml) to the system and incubated at 55 for 1 hr. The beads were then removed and DNA fragments in the supernatants were extracted using MinElute PCR Purification Kit (28004, Qiagen).

Quantitive Real time PCR were performed with primers:
*Zfp366*-primer1: (forward) 5'-CTTTGGATCGGGACTGGACC-3',
(reverse) 5'-TGAATGGGGGAGCCATAGGGA-3';
*Zfp366*-primer2: (forward) 5'-GTGCTGGGGTTTCAAAGCAG-3',
(reverse) 5'-GGGCTAACCAAGAGGGAACC-3';
*Zfp366*-primer3: (forward) 5'-GCCACCACAAAGAACCACAC-3',
(reverse) 5'-GAGCTGGGCCCAAATCATCT-3';
*Zbtb46* primer1: (forward) 5'- GAGGAACAAGGTAGCCCCAG-3',
(reverse) 5'-CCCATACACCACTTGCCCTT-3';
*Zbtb46* primer2: (forward) 5'-TCACATCTGGGTGGGATTGC-3',
(reverse) 5'-TCCGTTGCTGTCACGGTTTA-3';
*Zbtb46* primer3: (forward) 5'-CTTGCCTAGCACCCAGCTAA-3',
(reverse) 5'-GCAAACCCATCCAATGCTCC-3';
*Batf3* primer1: (forward) 5'-TGCACAGCAAGTTCTAGCGA-3',
(reverse) 5'-TCCCCAAACCAACGTTCACA-3';
*Batf3* primer2: (forward) 5'-GGGGCAGAAGTTTGTGAACG-3',
(reverse) 5'-GTCAGGCCCTGTGTTCCATA-3'.

## Statistical analysis

Analysis of all data was done with unpaired two-tailed Student's *t*-test (Prism, GraphPad Software). p<0.05 was considered significant. *p<0.05; **p<0.01; ***p<0.001; ****p<0.0001.

## Acknowledgements

We thank Prof. Fang-Lin Sun (Tongji University, China) for providing B6N-Tyr<sup>c-Brd</sup>Hdac3<sup>tm1a(EUCOMM)Wtsi</sup>/Wtsi mice and Prof. Nan Shen (Shanghai Jiao Tong University, China) for providing Itgax-Cre mice. We are grateful for the support provided by the animal core facility at Tsinghua University. This research was supported by the Ministry of Science and Technology of China (National Key Research Projects 2019YFA0508502 to L Wu, 2022YFC2505001 to L Wu), the National Natural Science Foundation of China (grants 31991174 to L Wu, 31800769 to Z He), the National Basic Research Center of China (82388101 to L Wu), funding and funding from Tsinghua-Peking Center for Life Sciences (to L Wu).

## Additional information

### Funding

| Funder | Grant reference number | Author |
| --- | --- | --- |
| Ministry of Science and Technology of the People's Republic of China | National Key Research Project 2019YFA0508502 | Li Wu |
| National Natural Science Foundation of China | 31991174 | Li Wu |
| National Natural Science Foundation of China | 31800769 | Zhimin He |
| Tsinghua-Peking Center for Life Sciences | | Li Wu |
| Ministry of Science and Technology of the People's Republic of China | National Key Research Project 2022YFC2505001 | Li Wu |
| National Basic Research Center of China | 82388101 | Li Wu |

The funders had no role in study design, data collection and interpretation, or the decision to submit the work for publication.

### Author contributions

Yijun Zhang, Conceptualization, Data curation, Formal analysis, Investigation, Methodology, Writing - original draft; Tao Wu, Data curation, Formal analysis, Investigation, Methodology, Writing – review and editing; Zhimin He, Data curation, Formal analysis, Funding acquisition, Investigation, Methodology, Writing – review and editing; Wenlong Lai, Conceptualization, Investigation, Methodology; Xiangyi Shen, Investigation, Methodology; Jiaoyan Lv, Methodology, Writing – review and editing; Yuanhao Wang, Data curation, Validation; Li Wu, Conceptualization, Resources, Data curation, Formal analysis, Supervision, Funding acquisition, Methodology, Project administration, Writing – review and editing

### Author ORCIDs

Tao Wu ⓘ http://orcid.org/0000-0001-9347-5723
Xiangyi Shen ⓘ http://orcid.org/0000-0002-7460-1066
Li Wu ⓘ https://orcid.org/0000-0002-2802-1220

### Ethics

All animal procedures were performed in strict accordance with the recommendations and approval of the Institutional Animal Care and Use Committee of Tsinghua University, protocol permit number: 18-WL2.

### Decision letter and Author response

Decision letter https://doi.org/10.7554/eLife.80477.sa1
Author response https://doi.org/10.7554/eLife.80477.sa2

## Additional files

### Supplementary files
• Supplementary file 1. Differentially expressed genes in Hdac3 deficient bone marrow pDCs. Genes increased or decreased in Hdac3 deficient bone pDCs (fold change ≥2 and *P*-value ≤0.5).

• MDAR checklist

### Data availability
For original data, please contact wuli@mail.tsinghua.edu.cn RNA sequencing data are available at NCBI GEO Datasets under accession GSE197207, methods were described. Cut & Tag data are available at NCBI GEO Datasets under accession GSE197212, methods were described.

The following datasets were generated:

| Author(s) | Year | Dataset title | Dataset URL | Database and Identifier |
|---|---|---|---|---|
| Wu L, Zhang Y, He Z | 2022 | Genome-wide maps of H3K27ac in HDAC3 knockout bone marrow pDCs compared with control pDCs | https://www.ncbi.nlm.nih.gov/geo/query/acc.cgi?acc=GSE197212 | NCBI Gene Expression Omnibus, GSE197212 |
| Wu L, Zhang Y, He Z | 2022 | Next Generation Sequencing of transcriptomes to reveal the role of HDAC3 in regulating the development of LMPP and CD115- CDP and bone marrow pDCs | https://www.ncbi.nlm.nih.gov/geo/query/acc.cgi?acc=GSE197207 | NCBI Gene Expression Omnibus, GSE197207 |

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
