## [Editor Report]

This valuable study examines the expression of the histone-modifying enzyme HDAC3 within the dendritic cell (DC) compartment by taking advantage of a tamoxifen-inducible ERT2-cre mouse model to report the dependency of plasmacytoid DCs (pDCs) but not conventional DC (cDCs)s on HDAC3 at the common lymphoid progenitor stage during DC development. The methods are convincing and include RNA seq studies that identify multiple DC-specific target genes within the remaining pDCs, and the use of Cut and Tag technology to validate some of the identified targets of HDAC3. Taken together, this study shows the requirement of HDAC3 for pDCs but not for cDCs, congruent with the recent findings of a lymphoid origin of pDCs, and will be of great interest to immunologists.

---

## [Decision Letter]

**Decision letter after peer review:**

Thank you for submitting your article "Regulation of pDC Fate Determination by Histone Deacetylase 3" for consideration by *eLife*. Your article has been reviewed by 2 peer reviewers, and the evaluation has been overseen by a Reviewing Editor and Satyajit Rath as the Senior Editor. The following individual involved in review of your submission has agreed to reveal their identity: Roxane Tussiwand (Reviewer #1).

Essential revisions:

*Reviewer #1 (Recommendations for the authors):*

Specific points:

In Figure 2 the authors observe a relative increase of cDC and a decrease of pDCs. It is important to define the pre-gating strategy. To determine if there the decrease of pDCs reflects on the cDC increase, a mixed BM chimeras would be essential, such that both subsets can be compared in competitive settings. For this purpose the use of CD45.1-Cd45.2 and the intercross of CD45.1.2 heterozygous mouse might be suitable. (regarding the Mixed BM chimera done in Figure 4 please see comment below).

The reasoning that the defect is cell intrinsic is imprecise. The fact that HDAC3 is expressed on HSCs and CLPs could reflect a defect that is already present within the progenitor compartment. In light of the recent evidence that pDCs are mostly generated from lymphoid progenitors in BM, providing data for chimera experiments on B, T and NK cell reconstitution as well as a more detailed progenitor subset analysis would be helpful to determine at which stage the defect is most prominent.

Figure 3 LSKs are increased in numbers. Each pool of cells is defined by the proliferation, differentiation and apoptosis rate. Therefore, all three parameters must be analyzed to define the reason for expansion or reduction. i.e. are HDAC cKO precursors undergoing increased proliferation? Is this a consequence of more cytokines availability due to the lack of other subsets, such as B cells? To achieve differentiation cells need to reduce proliferation and upregulate lineage specific programs. Is the increased abundance of stem cells related to the incapacity of downregulating proliferation and could this be the reason for impaired differentiation? The analysis of cell cycle stages within the progenitor compartment may be useful to address this aspect. In parallel to here performed BrDU pulse experiment a pulse and chase or a chase experiment may help further clarify. It will be nevertheless important to dissect all subsets more carefully given the broad expression of HDC3 across precursors

A more recent classification of progenitors could be used for HSCs and DC-progenitors:

- "Pietras et al. 2015" (MPP2 MPP3 and MPP4) and "Klein et al. 2022" Lymphoid committed (ESAM+ and ESAM- MPP4)

- cDC progenitors "Durai et al. 2020 (CD226-preDC1)"

- pre DC2 "Grajales-Reyes GE 2015" and "Schlitzer A. 2015"

Also analysis of competitive chimeras may help in dissecting the developmental defect resulting from HDAC3 deficiency. It is well known that Flt3L is the instructive cytokine for DCs and pDC development.

FLt3 is highly expressed on Lymphoid progenitors that comprise pDC precursors, while cDC progenitors are defined by the expression of CSF1R,a s correctly stated by the authors. Given that after tamoxifen treatment, Flt3 expression is almost completely absent on lymphoid progenitors it is likely that the cells are incapable to respond to Flt3-Flt3L instructive signal. Transduction of BM progenitors with Flt3R may allow to address this point.

Figure 3E. the gate shown will include auto florescent cells please use a better gating strategy for identifying CLPs.i.e include an auto florescent exclusion gate, see comment above on looking at MPP4s.

In Figure 4C CD115-CDPs are transplanted, this subset is not generating the necessary cells for the survival of lethally irradiated mice and indeed the reconstitution percentage is also for WT cells below 10%, suggesting that recipient mice are not lethally irradiated to allow for their survival and contribute to the hematopoietic compartment. Could you. Please specify if in this context mice were lethally or sublethally irradiated.

To avoid the confusion between recipient derived cells and donor derived WT cells I would suggest using, as mentioned above, CD45.1.2 heterozygous mice as WT donors or recipients.

The fact that HDAC3 deficiency results in a significant disadvantage of HSC reconstitution suggest that the defect is upstream any DC/pDC lineage specification. What is the cDC reconstitution in HSC transplants (Figure 4E)? In the absence of HDAC3 what cells develop?

To claim that the defect is pDC specific a better Cre depleting mouse model would be advisable, and indeed the authors have generated the CD11c cre intercross, which however did not result in a pDC developmental defect. Where this also analyzed in a competitive setting? SiglecH cre model exist, however I do not know whether it would be an accessible model for the authors.

In Figure 6 it appears that HDAC3 inhibition is impairing not only pDCs but also cDCs and total cells numbers, rather suggesting that there is an effect on viability or proliferation or both at higher concentrations. How specific is this inhibitor on HDAC3? This result would argue against a specific pDC effect, and the statement (Page 10 line 257) "This result suggested that HDAC3 deacetylase activity was required preferentially for pDC development" should be adjusted.

To Figure 7

Could you provide with the full tables of genes that are deregulated? What are the most highly expressed genes in cKO cells? Please provide also an unbiased analysis of the data, i.e. findMarkers….

For the development of pDCs two transcription factors were shown to be essential Tcf4 and Irf8. It would therefore be important to show among the selected TFs also these two.

---

## [Author Response]

Essential revisions:Reviewer #1 (Recommendations for the authors):Specific points:In Figure 2 the authors observe a relative increase of cDC and a decrease of pDCs. It is important to define the pre-gating strategy. To determine if there the decrease of pDCs reflects on the cDC increase, a mixed BM chimeras would be essential, such that both subsets can be compared in competitive settings. For this purpose the use of CD45.1-Cd45.2 and the intercross of CD45.1.2 heterozygous mouse might be suitable. (regarding the Mixed BM chimera done in Figure 4 please see comment below).

We thank the reviewer for the suggestion. In this study we defined pDCs and cDCs using the same pre-gating (Figure 4 and Figure 5). When the data from three independent experiments were pooled, we found no significant changes in the percentage of cDCs in cKO mice. As optimized in Figure 2B and 2C in revised manuscript. The relevant description has been added and underlined in the revised manuscript Page 6 Line 148-149.

We agree that utilizing BM cells from CD45.1.2 mice as WT donors or recipients would enable us to effectively distinguish the origin of progeny cells. However, in this particular experiment, we opted to construct the competitive BM chimeric mice by using CD45.1 mice as recipients, the CD45.2 mice (Hdac3-Ctrl and Hdac3-cKO) as donors, and the CD45.1 wild-type BM cells were used as competitors. Using this specific combination we could clearly distinguish the cells derived from CD45.2 donor BM cells within the chimeric mice, which allowed us to effectively compare the differentiation capacity of Hdac3-Ctrl and Hdac3-KO BM cells within the same competitive environment.

The reasoning that the defect is cell intrinsic is imprecise. The fact that HDAC3 is expressed on HSCs and CLPs could reflect a defect that is already present within the progenitor compartment. In light of the recent evidence that pDCs are mostly generated from lymphoid progenitors in BM, providing data for chimera experiments on B, T and NK cell reconstitution as well as a more detailed progenitor subset analysis would be helpful to determine at which stage the defect is most prominent.

We totally agree that the statement “the defect is cell intrinsic” is imprecise. We actually wanted to illustrate that HDAC3 could regulate pDC development in a cell-intrinsic way in the early stage, but did not mean that the pDC defects in HDAC3 mice were pDC cell intrinsic. The relevant description has been underlined in the revised manuscript Page 6 Line 153-154.

In this study, we found no significant changes in T cells, mature B cells or NK cells, but immature B cells were dramatically decreased, in Hdac3- ERT2-Cre mice after tamoxifen treatment (Figure 6). However, in the bone marrow chimera experiment, numbers of major lymphoid cells were decreased due to the impaired reconstitution capacity of Hdac3 deficient progenitors, as shown in Figure 2A-C in revised manuscript. Additionally, HDAC3 has been shown to be required for T cell and B cell generation, in HDAC3-VavCre mice (Summers et al., 2013), which was more likely due to the defect of early lymphoid progenitors.

We have observed significant impairment of pDCs in the tamoxifen-treated *Hdac3^f/f^ CreER^T2^* mice, while there were no notable changes in *Hdac3*^f/f^*-Itgax cre* mice, as shown in Figure 5−figure supplement 1C-D in revised manuscript. Additionally, *Hdac3*-deficient progenitors, including HSCs, CMPs, CLPs, and CDPs, exhibited defects in pDC generation (Figure 5 in revised manuscript), suggesting that HDAC3 may be involved in determining pDC generation at the CLP and/or CDP stages.

Figure 3 LSKs are increased in numbers. Each pool of cells is defined by the proliferation, differentiation and apoptosis rate. Therefore, all three parameters must be analyzed to define the reason for expansion or reduction. i.e. are HDAC cKO precursors undergoing increased proliferation? Is this a consequence of more cytokines availability due to the lack of other subsets, such as B cells? To achieve differentiation cells need to reduce proliferation and upregulate lineage specific programs. Is the increased abundance of stem cells related to the incapacity of downregulating proliferation and could this be the reason for impaired differentiation? The analysis of cell cycle stages within the progenitor compartment may be useful to address this aspect. In parallel to here performed BrDU pulse experiment a pulse and chase or a chase experiment may help further clarify. It will be nevertheless important to dissect all subsets more carefully given the broad expression of HDC3 across precursors

We thank the reviewer for the detailed comments and constructive suggestions. The issues concerned by the reviewer have been well illustrated in the published work (Summers et al., 2013), where they have confirmed that HDAC3 is required for HSC homeostasis by regulating DNA replication. Consistently, our observations in BrdU and cell cycle analysis also demonstrated an increase in the number and proliferation rate of HSCs in *Hdac3* deficient mice (Figure 7), with a notable accumulation at the S phase of the cell cycle (Figure 8), and the increased cell number were mainly within MPP2 and MPP3., the progenitors for myeloid lineage differentiation.

A more recent classification of progenitors could be used for HSCs and DC-progenitors:- "Pietras et al. 2015" (MPP2 MPP3 and MPP4) and "Klein et al. 2022" Lymphoid committed (ESAM+ and ESAM- MPP4)- cDC progenitors "Durai et al. 2020 (CD226-preDC1)"- pre DC2 "Grajales-Reyes GE 2015" and "Schlitzer A. 2015"

We greatly appreciate the highly useful references recommended by the reviewer. Accordingly, we re-analyzed the hematopoietic stem/progenitor cells by using the gating strategy described by "Pietras et al. 2015”. The LT-HSC, ST-HSC, MPP2, MPP3, MPP4 and CLPs among lineage negative cells in bone marrow were gated separately. We found that LSK cells were increased in *Hdac3* deficient mice, especially the MPP2 and MPP3, whereas, no significant changes in MPP4 were observed. In contrast, the numbers of LT-HSC and ST-HSC and CLP were all dramatically decreased (Figure 3). These results have been integrated in *Figure 3A-C* in revised manuscript. The relevant description has been added and underlined in the revised manuscript Page 6 Line 164-168, Page 7 Line 180.

We also analyzed pre-cDC1, based on the recommended gating strategy (Durai et al. 2020) and observed a significant decrease in the number of bone marrow pre-cDC1 in *Hdac3* deficient mice (Figure 9). These results for pre-cDC1 were added into Figure 3−figure supplement 1D-F in revised manuscript. The relevant description has been added and underlined in the revised manuscript Page 7 Line 180-183.

According to the reviewer’s suggestion, we also analyzed the pre-cDC subpopulations, based on the recommended gating strategy (Schlitzer et al., 2015). In this analysis, a significant decrease was observed in the number of total population of pre-DCs, including all four major subpopulations (Figure 10 below). According to the published work (Schlitzer et al., 2015), where they described the different subpopulations of pre-DCs, Siglec H^+^Ly6C^−^ (R3) represents the early pre-DCs that can differentiate into pDCs, cDC1s, and cDC2s. Siglec H^+^Ly6C^+^ (R2) and Siglec H^−^Ly6C^+^ (R1) were identified as cDC2-primed progenitors, while Siglec H^−^Ly6C^−^ (R4) represents the cDC1-primed progenitor population, which represents the major remaining pre-DC population in *Hdac3* deficient mice. The results of analysis of pre-DC were added to Figure3−figure supplement 1A-C. The relevant description has been added in the text (underlined, Page 7 Line 180-183) of revised manuscript.

Also analysis of competitive chimeras may help in dissecting the developmental defect resulting from HDAC3 deficiency. It is well known that Flt3L is the instructive cytokine for DCs and pDC development.FLt3 is highly expressed on Lymphoid progenitors that comprise pDC precursors, while cDC progenitors are defined by the expression of CSF1R,a s correctly stated by the authors. Given that after tamoxifen treatment, Flt3 expression is almost completely absent on lymphoid progenitors it is likely that the cells are incapable to respond to Flt3-Flt3L instructive signal. Transduction of BM progenitors with Flt3R may allow to address this point.

As concerned by the reviewer, our data showed that the Flt3 expression was significantly decreased on LT-HSCs and CMPs, but comparable on ST-HSCs and MPPs, as well as the pDCs and cDC subsets (Figure 11 and Figure 12), indicating that Flt3 was not completely absent. Moreover, Flt3 signaling is essential for the development of both pDCs and cDCs, the fact that the cDCs could still be generated, although in reduced numbers, in HDAC3 deficient mice suggested that the expression of Flt3 might not be the major causative factor for the defective phenotype of pDC in *Hdac3* deficient mice.

Figure 3E. the gate shown will include auto florescent cells please use a better gating strategy for identifying CLPs.i.e include an auto florescent exclusion gate, see comment above on looking at MPP4s.

We thank the reviewer’s kind suggestion. The gating strategy for CLP was optimized and auto florescent cells were excluded (Figure 2 and Figure 3), as shown in Figure 3A of the revised manuscript.

In Figure 4C CD115-CDPs are transplanted, this subset is not generating the necessary cells for the survival of lethally irradiated mice and indeed the reconstitution percentage is also for WT cells below 10%, suggesting that recipient mice are not lethally irradiated to allow for their survival and contribute to the hematopoietic compartment. Could you. Please specify if in this context mice were lethally or sublethally irradiated.

In CD115^−^CDP transplantation experiments, we co-transferred CD115^−^CDPs (CD45.2) isolated from HDAC3 deficient or WT control mice together with WT total bone marrow cells (CD45.1) into lethally irradiated CD45.1 recipient mice. In this case the co-transferred CD45.1 WT bone marrow cells could support the survival of the recipient mice, and consequently, the proportions of HDAC3 deficient CD115^−^CDP-derived cells (CD45.2) and the WT CD115^−^CDP-derived cells in the BM and spleens of recipient mice were quite low, typically below 10%.

To avoid the confusion between recipient derived cells and donor derived WT cells I would suggest using, as mentioned above, CD45.1.2 heterozygous mice as WT donors or recipients.

In this study, we used CD45.1 mice as both competitor donors as well the recipients, and CD45.2 mice (either Hdac3-Ctrl or Hdac3-cKO) as donors respectively. In this setting, we could compare the differentiation capacity between CD45.2 Hdac3-Ctrl with that of CD45.2 Hdac3-cKO BM cells. This competitive setting was not for the purpose of comparing the progenies of the CD45.1 competitor cells with that of CD45.1 recipients. We apologize for not describing the purpose of these experiments clearly in the manuscript.

The fact that HDAC3 deficiency results in a significant disadvantage of HSC reconstitution suggest that the defect is upstream any DC/pDC lineage specification. What is the cDC reconstitution in HSC transplants (Figure 4E)? In the absence of HDAC3 what cells develop?

We agree with the reviewer, *Hdac3* deficiency impacted the early stage of cDC/pDC lineage specification. *Hdac3* deficiency resulted in a significant reduction in the reconstitution capacity of LSKs. Consistently, in addition to the near complete absence of pDC, the number of cDCs derived from Hdac3 deficient LSKs was also dramatically decreased in the recipient mice. Similar changes were also observed in the in vitro DC differentiation assay of LSKs.

Furthermore, we detected major lymphoid and myeloid cells in bone marrow in mice after short-term deletion of Hdac3 by tamoxifen (about one weeks after the final time of tamoxifen treatment), T cells, NK cells were not changed, monocytes and neutrophils were decreased, while macrophage and eosinophils were increased in cell number. However, in Hdac3-deficient HSC transfer experiment, the numbers of total progeny cells were dramatically decreased, as shown in Figure 2B in revised manuscript.

To claim that the defect is pDC specific a better Cre depleting mouse model would be advisable, and indeed the authors have generated the CD11c cre intercross, which however did not result in a pDC developmental defect. Where this also analyzed in a competitive setting? SiglecH cre model exist, however I do not know whether it would be an accessible model for the authors.

We agree that Siglec H cre mouse model would be a valuable tool to investigate the specific role of HDAC3 in pDCs differentiation, but this mouse model is not yet readily available. Meanwhile, as in Figure 5—figure supplement 1, we found that despite the efficient deletion of HDAC3 in pDCs of HDAC3^fl/fl^-CD11c cre mice, no significant developmental impairment in pDCs was observed, which demonstrated that HDAC3 was dispensable for the late stages of pDC development or pDC maintenance.

In Figure 6 it appears that HDAC3 inhibition is impairing not only pDCs but also cDCs and total cells numbers, rather suggesting that there is an effect on viability or proliferation or both at higher concentrations. How specific is this inhibitor on HDAC3? This result would argue against a specific pDC effect, and the statement (Page 10 line 257) "This result suggested that HDAC3 deacetylase activity was required preferentially for pDC development" should be adjusted.

In this study, we used RGFP966, which has been confirmed to be a highly specific HDAC3 inhibitor widely used in the field of epigenetic research (Malvaez et al., 2013). We conducted experiments using a range of concentrations of RGFP966 in in vitro DC differentiation assays. At the low concentration (0.2μM), a significant decrease in pDCs number was observed, while the number of cDCs remained largely unaffected. As the concentration increased, the numbers of both pDCs and cDCs decreased or nearly absent, which is more likely due to the cytotoxic effects of RGFP966 at higher concentrations (0.5μM and 1.0μM). These results suggested that at the optimal concentration of the inhibitor, pDC development was more reliant on HDAC3 compared to cDCs.

To Figure 7Could you provide with the full tables of genes that are deregulated? What are the most highly expressed genes in cKO cells? Please provide also an unbiased analysis of the data, i.e. findMarkers….

We have added the complete table comprising the significantly deregulated genes in the supplementary materials (Table 1). We selected the dramatically increased genes in the Hdac3 deficient pDCs for further gene enrichment analysis. We noticed that most of these genes were involved in cell-cycle and cell proliferation process (Figure 13 below), consistent with the increased number of LSKs in Hdac3 deficient mice. This result has been optimized and added as Figure 7—figure supplement 1 in revised manuscript. The relevant description has been added and underlined in the revised manuscript Page 11 Line 289-290.

For the development of pDCs two transcription factors were shown to be essential Tcf4 and Irf8. It would therefore be important to show among the selected TFs also these two.

We agree with the reviewer’s points. We have observed a slight decrease in both *Tcf4* and *Irf8*, as indicated by the RNA-seq analysis as shown in Figure 14, (this information is added in Figure7−figure supplement 1). Furthermore, we have confirmed the decrease of *Tcf4* through qRT-PCR analysis, as demonstrated in Figure 7B in the revised manuscript.